# Evolutionary–developmental (evo-devo) dynamics of hominin brain size

**Mauricio González-Forero** ⬤ ✉

Brain size tripled in the human lineage over four million years, but why this occurred remains uncertain. Here, to study what caused this brain expansion, I mathematically model the evolutionary and developmental (evo-devo) dynamics of hominin brain size. The model recovers (1) the evolution of brain and body sizes of seven hominin species starting from brain and body sizes of the australopithecine scale, (2) the evolution of the hominin brain–body allometry and (3) major patterns of human development and evolution. I show that the brain expansion recovered is not caused by direct selection for brain size but by its genetic correlation with developmentally late preovulatory ovarian follicles. This correlation is generated over development if individuals experience a challenging ecology and seemingly cumulative culture, among other conditions. These findings show that the evolution of exceptionally adaptive traits may not be primarily caused by selection for them but by developmental constraints that divert selection.

The human brain provides hardware for stunning achievements, but why it evolved remains unresolved. The fossil record shows a sharp expansion in hominin brain size, tripling over the past four million years from australopithecines to modern humans[1], while some *Homo* were small-brained[2,3]. Many hypotheses exist for why such hominin brain expansion occurred[4–22] and they are actively tested, often with correlative[23–25] or comparative studies in non-hominin species[26–29]. Yet, establishing what were the causes of hominin brain expansion remains a major multidisciplinary challenge.

A promising but underexploited approach to identifying the causes of hominin brain expansion is by means of mechanistic modelling. While causes can be inferred from intervention effects[30], interventions are often infeasible or impractical for studying hominin brain expansion. Yet, models that mechanistically replicate an event of interest allow for probing the event's underlying causes through simulated interventions[31]. Models of brain evolution often provide qualitative insights into conditions conducive to large brain evolution[32–35], but it is key that models make quantitative predictions for the evolution of human-sized brains (for example, approximately 1.3 kg for an average adult female). This distinction is crucial, as factors favouring brain expansion may not lead to a human-sized brain but to brains that may be either too small or too large relative to those of humans.

A recent mathematical model—hereafter, the brain model—makes quantitative predictions for conditions under which a given brain size evolves[36]. The brain model mechanistically replicates the evolution of adult brain and body sizes of six *Homo* species and much of the timing of human development, including the length of childhood, adolescence and adulthood[37]. Analysis of the brain model[37] has found causal, computational evidence that a challenging ecology[7,15,22] and seemingly cumulative culture[14,19,21] rather than social interactions[6,9,12,16] could have caused hominin brain expansion. In the model, a challenging ecology, where individuals need brain-supported skills to obtain energy, promotes brain expansion[36]. If, in addition, learning has weakly, not strongly, diminishing returns, then human-sized brains and bodies can evolve[37]. Although the model does not explicitly model cultural dynamics, weakly diminishing returns of learning could in principle arise from culture if skilled individuals can keep learning from accumulated knowledge in the population[37]. Thus, in the model, hominin brain expansion needs both a challenging ecology and seemingly cumulative culture, presumably to reap the benefits in adulthood of investing in growing large brains during childhood. By contrast, conflicting interests between social partners enable evolutionary arms races in brain size as proposed by influential hypotheses[6,9,16], but the arms races fail to yield evolutionarily stable human-sized brains and

School of Biology, University of St Andrews, St Andrews, UK. ✉e-mail: mgf3@st-andrews.ac.uk

## Table 1 | Key parameters

| Parameter | Value | Interpretation |
|---|---|---|
| Brain maintenance cost[a], $B_b$ | 313 MJ kg$^{-1}$ yr$^{-1}$ | Mid |
| Follicle maintenance cost[b], $B_r$ | 2,697 MJ kg$^{-1}$ yr$^{-1}$ | High |
| Soma maintenance cost[a], $B_s$ | 30 MJ kg$^{-1}$ yr$^{-1}$ | Low |
| Memory cost[b], $B_k$ | 50 MJ TB$^{-1}$ yr$^{-1}$ | Mildly high |
| Learning cost, $E_k$ | 250 MJ TB$^{-1}$ | Mildly low |
| Brain allocation to skill, $s_k$ | 0.5 | Mid |
| Newborn skill, $x_{k1}$ | 0 TB | Low |
| Environmental difficulty, $a$ | 1.15 | Mildly high |
| Skill effectiveness, $\gamma$ | 0.6 TB$^{-1}$ | Mildly low |
| Competence, $c$(skill) | Exponential | Weakly DRL |

Human-scale brains and bodies evolve under these parameter values in the brain model. Changing one of these parameters at a time may substantially change the evolved brain or body sizes or their ontogenetic growth, even causing the evolutionary collapse of brain size (Figs. 6 and R in ref. 36 and Extended Data Fig. 3 in ref. 37; the interpretation in the third column is informed by the ranges that yield evolution of non-zero brain sizes). DRL, diminishing returns of learning. [a]Estimated from empirical data[39,98]. [b]Empirically informed[80,99].

bodies given their metabolic costs[37]. In turn, cooperation[12] disfavours brain size evolution as individuals can rely on social partners' brains to overcome ecological challenges and so can avoid investing in growing an expensive brain[37]. The model has incorporated basic aspects of leading hypotheses without explicitly modelling every aspect such as information manipulation or relationship management[12]. Yet, doing so has not been necessary to obtain the evolution of human-sized brains and bodies given the data used for parameter values.

The brain model makes quantitative predictions by explicitly considering development, that is, the construction of the phenotype over life. The model describes the construction of brain and body sizes over life using energy conservation analysis following ref. 38, which obtains an equation describing the developmental dynamics of body size depending on parameters measuring metabolic costs that can be easily estimated from data[38]. The brain model implements this approach to obtain equations describing the developmental dynamics of brain, reproductive and somatic tissue sizes depending on genotypic traits controlling energy allocation to the production of each tissue at each age[36]. For simplicity, reproductive tissue is defined in the model as preovulatory ovarian follicles that determine fertility, given that the model considers only females. The developmentally dynamic equations define the developmental constraints, as the phenotype is constrained to satisfy such equations. The brain model thus depends on parameters measuring brain metabolic costs, which are thought to be a key reason not to evolve large brains[11,17] and which are easily estimated from existing data[39]. In the model, the genotypic traits evolve, which leads to the evolution of brain and body sizes in kilograms, whose units arise from the empirically estimated metabolic costs. The model has identified key parameters that have strong effects on brain size evolution and particular parameter values that enable the evolution of human-scale brains and bodies[36,37] (Table 1).

However, further understanding from the brain model has been hindered by the long-standing lack of mathematical synthesis between development and evolution[40–42]. To consider developmental dynamics, the brain model was evolutionarily static: it had to assume evolutionary equilibrium where the evolved genotypic traits are optimal in that they maximize fitness and so, since the model considers developmental dynamics, it was analysed using dynamic optimization, specifically optimal control theory, as is standard in life history theory[43–46]. This was done because of the long-standing lack of mathematical integration of development and evolution, which meant that there were no tractable methods to mathematically model the evolutionary and developmental (evo-devo) dynamics of the brain model. Approaches

available at the time that mathematically integrated developmental and evolutionary dynamics required computation of functional derivatives and solution of integro-differential equations[47,48], both of which are prohibitively challenging for the relatively complex brain model. Yet, considering the evolutionary dynamics could yield richer insight. For instance, a debated topic concerns the roles of selection and constraint in brain evolution, often studied with correlational approaches[49–53]. Considering the evolutionary dynamics in the brain model could enable causal analyses of these roles in hominin brain expansion. Indeed, the short-term evolutionary dynamics can be described as the product of direct selection and genetic covariation, assuming negligible genetic evolution[54,55], where genetic covariation is a key descriptor of evolutionary constraints[54,56,57]. Using this separation, a previous study found that selection for brain size must have driven brain and body size increases from *Australopithecus afarensis* to *Homo sapiens*, assuming that selection for other traits is not relevant and that genetic covariation is constant over long periods[58]. Yet, the lack of mathematical integration of developmental and evolutionary dynamics has meant that there is a lack of tools to separate selection from constraint in the brain model and in long-term evolution, without assuming negligible genetic evolution.

A solution to these difficulties is offered by a recent mathematical framework—hereafter, evo-devo dynamics framework—that integrates evo-devo dynamics, allowing for mathematically modelling the evo-devo dynamics for a broad class of models, assuming clonal reproduction and rare, weak and unbiased mutation[59]. This framework provides equations that separate the effects of selection and constraint for long-term evolution under non-negligible genetic evolution and evolving genetic covariation. Moreover, the framework provides equations to analyse evolutionary aspects in developmentally explicit models such as the brain model, including what is under selection in the model, how brain metabolic costs translate into fitness costs and how brain size development translates into genetic covariation.

In this Article, to gain a deeper understanding of why hominin brain expansion could have occurred, I implement the brain model[37] in the evo-devo dynamics framework[59]. This yields a model of the evo-devo dynamics of hominin brain size that mechanistically recovers in silico the hominin brain expansion from australopithecines to modern humans and multiple observations of human evolution and development. This evo-devo dynamics approach enables deeper analysis, showing that hominin brain expansion occurs in the model because of direct selection on follicle count rather than on brain size (Extended Data Fig. 1). The brain expands in the model because ecology and possibly culture make brain size and developmentally late follicle count 'mechanistically socio-genetically' correlated. This notion is similar to the classic notion of genetic covariation in quantitative genetics but differs in two aspects. First, 'mechanistic' genetic covariation arises from a mechanistic description of development rather than from a regression-based description as in quantitative genetics, which allows one to model long-term rather than only short-term phenotypic evolution[59,60]. Second, 'socio-genetic' covariation includes a mechanistic description of indirect genetic effects[61] and considers not only heredity but also the stabilization (or legacy) of the phenotype owing to social development, where phenotype construction depends on social partners[59,60]. Social development in the brain model occurs because cooperation and competition for energy extraction affect development. This mechanistic treatment shows that brain metabolic costs in the model are not direct fitness costs but affect mechanistic socio-genetic covariation, and that the evolutionary role of ecology and culture in the recovered hominin brain expansion is not to affect direct fitness costs or benefits but to generate the socio-genetic covariation that causes brain expansion.

I provide an overview of the model in Methods. I describe the model in detail and derive the necessary equations for the evo-devo analysis in Supplementary Information. I provide in Supplementary

Information the computer code[62] written in the freely accessible and computationally fast Julia programming language[63].

## Results

### Evolution of brain and body sizes of seven hominins

In the brain model, each individual obtains energy by using her skills to overcome energy-extraction challenges that can be of four types: ecological (for example, foraging alone), cooperative (for example, foraging with a peer), between-individual competitive (for example, outsmarting a peer) and between-group competitive (for example, two peers outsmarting two peers). The probability of facing a challenge of type $j$ at a given age is $P_j$ ($\sum_{j=1}^{4} P_j = 1$, where $j \in \{1, ..., 4\}$ indexes the respective challenge types). Assuming evolutionary equilibrium, the brain model was previously found to recover the evolution of the adult brain and body sizes of six *Homo* species and less accurately of *A. afarensis* by varying only the energy extraction time budget (EETB; the proportion of the different types of energy-extraction challenge faced) and the shape of the energy extraction efficiency (EEE) with respect to one's own or social partner's skills[37]. I recover these results with the evo-devo dynamics approach (Fig. 1). In these results, brain expansion from one evolutionary equilibrium to another is caused by an increasing proportion of ecological challenges and a switch from strongly to weakly diminishing returns of learning. As weakly diminishing returns of learning might arise from accumulated cultural knowledge in the population, this indicates that ecology and possibly culture cause hominin brain expansion in the model[37]. Below, I describe evo-devo patterns underlying such brain expansion and analyse further the factors causing it.

### Emergence of hominin brain–body allometry

To examine the influence of development alone on the developed brain and body sizes, I consider genotypic variation without evolution as follows. Consider the parameter values in the *sapiens* scenario of Fig. 1, which yield the evolution of brain and body sizes of *H. sapiens*. Under those parameter values and without evolution, randomly sampled genotypes develop adult brain and body sizes, generating a tight brain–body allometry with slope 0.54 ($R^2 = 0.95$; Fig. 2a). A similar slope but with a lower placement ('intercept') is found in other primates and mammals[64] (Fig. 2a). As there is only development but no evolution in Fig. 2a, this 0.54 slope arises purely from developmental canalization sensu ref. 65. For the sample size used, no organism with random genotype reaches hominin brain and body sizes (no black dot in the green region in Fig. 2a). The recovered brain–body allometry from developmental canalization has a high placement, so the developed brain size is relatively large for the developed body size. In simpler models of development, an allometry with high placement is known to arise with a growth rate, developmentally initial size or growth duration that is high for the predicted variable (here adult brain size) relative to the predictor (here adult body size)[66]. In Fig. 2a, brain size can have a high growth rate and growth duration because of the parameter values in the *sapiens* scenario, including a high proportion of difficult ecological challenges, weakly diminishing returns of learning and a high metabolic cost of memory (Figs. 3 and 6 in ref. 36 and Extended Data Fig. 1 in ref. 37). Hence, in the brain model under the *sapiens* scenario, development alone has a strong influence on the developed brain and body sizes, with a developmental bias[57] towards large brains, but is unlikely to yield hominin brain sizes without selection.

Letting evolution proceed, I find that the evolved brain and body sizes strongly depend on the ancestral genotypic traits. For instance, under the *sapiens* scenario, evolving human-sized brains requires that the ancestral genotypic traits develop large bodies; otherwise, brain size may collapse over evolution (Supplementary Fig. 1). Yet, ancestral genotypic traits developing a large body are not needed to evolve large brains when facing ecological challenges alone (Supplementary Fig. 2). The developmental patterns that evolve strongly

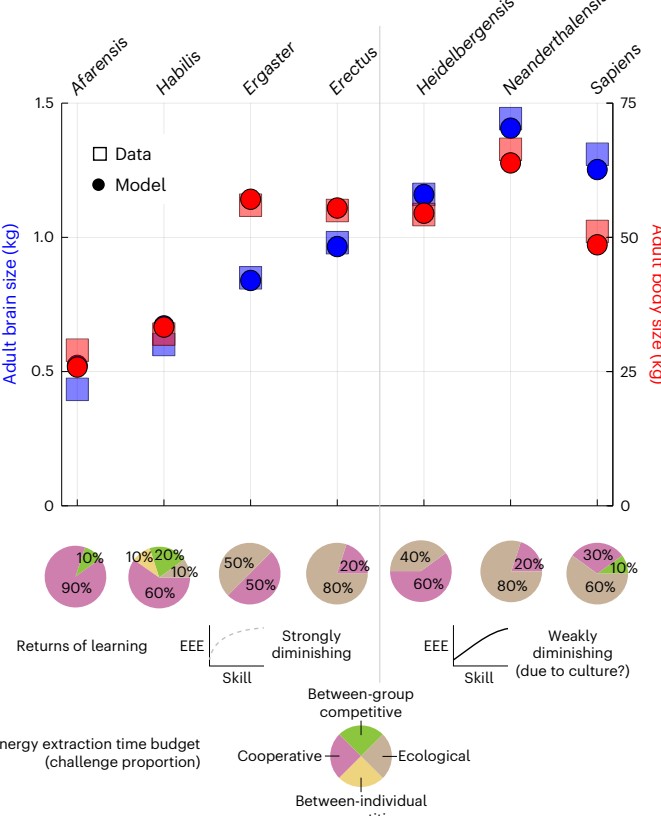

**Fig. 1 | Evolution of brain and body sizes of seven hominin species solely by changing socio-genetic covariation.** Adult brain and body sizes of seven hominin species evolve in the model only by changing the EETB, the returns of learning and how the skills of cooperating partners interact. Squares are the observed adult brain and body sizes for the species at the top (data from refs. 39,69,100–104). Dots are the evolved values in the model for a 40-year-old using the evo-devo dynamics approach. Pie charts give the EETB used in each scenario. The returns of learning are either strongly diminishing (power competence) for the left four scenarios or weakly diminishing (exponential competence) for the right three scenarios. Cooperation is either submultiplicative for the *afarensis* and right three scenarios or additive for the remaining scenarios. These EETBs and shapes of EEE were previously identified as evolving best-fitting adult brain and body sizes for the corresponding species, assuming evolutionary equilibrium[37]. In principle, weakly diminishing returns of learning might arise from culture. I will show that varying EETBs and the shape of EEE only varies socio-genetic covariation $\mathbf{L_z}$ but not the direction of direct selection $\partial w/\partial \mathbf{z}$ or where it is zero (it never is). I refer to the particular EETB and shape of EEE yielding the evolution of adult brain and body sizes of a given species as the species scenario. For the *afarensis* scenario, the ancestral genotypic traits are somewhatNaive2 (Supplementary equation (46)). For the remaining six scenarios, the ancestral genotypic traits are the final genotypic traits of the *afarensis* scenario started from the somewhatNaive2 genotypic traits. The final evolutionary time is 500 for all 7 scenarios. Pie charts reproduced with permission from ref. 37, Springer Nature Ltd.

depend on the ancestral genotypic traits, even if the evolved adult brain and body sizes are the same (Supplementary Fig. 3). The dependence of the evolved traits on ancestral conditions is sometimes called phylogenetic constraints, which are typically assumed to disappear with enough evolutionary time[67]. The evo-devo dynamics framework finds that phylogenetic constraints do not necessarily disappear with enough time as it finds that genetic constraints are necessarily absolute in long-term evolution[59]. This is because there is socio-genetic covariation only along the path where the developmental constraint is met (so $\mathbf{L_z}$ in equation (7), a mechanistic, generalized analogue of Lande's[54]

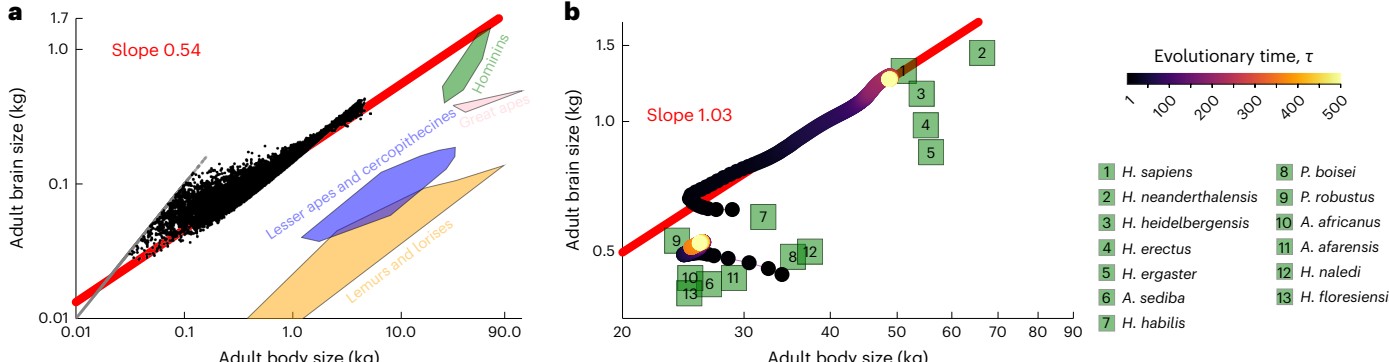

**Fig. 2 | Brain–body allometries without and with evolution. a**, Brain size at 40 years of age versus body size at 40 years of age on a log–log scale, developed under the brain model from $10^6$ randomly sampled genotypes (that is, growth efforts, drawn from the normal distribution with mean 0 and standard deviation 4) using the parameter values of the *sapiens* scenario. Black dots are 'non-failed' organisms, whose body is not entirely composed of brain at 40 years of age, and are approximately 4% of $10^6$. Grey dots are 'failed' organisms having small bodies (<200 g) at 40 years of age entirely composed of brain tissue owing to tissue decay from birth, and are about 96% of $10^6$ (Supplementary Fig. 12). Coloured regions encompass extant and fossil primate species. **b**, Brain size at 40 years of age versus body size at 40 years of age over evolutionary time on a log–log scale for two trajectories. The bottom trajectory uses the parameter values of the *afarensis* scenario (Fig. 1) and somewhatNaive2 ancestral genotypic traits. The top trajectory uses the parameter values of the *sapiens* scenario (Fig. 1) and the evolved genotypic traits of the bottom trajectory as ancestral genotypic traits. A linear regression over the top trajectory yields a slope of 1.03 (red line). Adult values for 13 hominin species are shown in green squares. Brain and body size data for non-hominins are from ref. 64, excluding three fossil, outlier cercopithecines; brain and body size data for hominins are from refs. 2,3,39,64,69,100–106 using only female data when possible. Fossil data may come from a single individual and body size estimates from fossils are subject to additional error. *H.*, *Homo*; *A.*, *Australopithecus*; *P.*, *Paranthropus*.

G matrix, is singular), which means that the evolutionary outcome depends on the evolutionarily initial conditions[59,68].

To identify suitable ancestral genotypic traits to model hominin brain expansion, I consider naive ancestral genotypic traits (termed somewhatNaive2) under the *afarensis* scenario (blue dots in Supplementary Fig. 4d–f). These ancestral genotypic traits cause individuals to develop brain and body sizes of australopithecine scale, most closely approaching those of *Paranthropus boisei* (initial evolutionary time of the bottom trajectory in Fig. 2b and blue dots in Extended Data Fig. 2a,c). With this ancestral genotype, I let evolution proceed under the *afarensis* scenario, which yields australopithecine brain and body sizes, most closely approaching those of *Paranthropus robustus* (bottom trajectory in Fig. 2b).

Setting the evolved genotypic traits under this *afarensis* scenario as ancestral genotypic traits and switching parameter values to the *sapiens* scenario yields an immediate plastic change in the developed brain and body sizes, approaching those seen in *habilis* (initial evolutionary time of the top trajectory in Fig. 2b). Letting evolution proceed yields the evolution of *H. sapiens* brain and body sizes (top trajectory in Fig. 2b). This evolutionary trajectory approaches the observed brain–body allometry in hominins starting from brain and body sizes of the australopithecine scale, with a slope of 1.03 (Fig. 2b).

The switch from the *afarensis* to the *sapiens* scenario involves a sharp decrease in cooperative challenges, a sharp increase in ecological challenges and a shift from strongly to weakly diminishing returns of learning (Fig. 1). While these changes are here implemented suddenly and so lead to an immediate plastic response, the changes may be gradual, allowing for genetic evolution.

**Evo-devo dynamics of brain size**

Further detail of the recovered hominin brain expansion is available by examining the evo-devo dynamics that underlie the *sapiens* trajectory in Fig. 2b. Such trajectory arises from the evolution of genotypic traits controlling energy allocation to growth. This evolution of energy allocation yields the following evo-devo dynamics in the phenotype.

Adult brain size more than doubles over evolution from around 0.6 kg to around 1.3 kg, closely approaching that observed in modern human females[39,69,70], and, over development, the evolved brain size displays several growth spurts (Fig. 3a).

Over evolution, follicle count (in mass units) shrinks early in life and expands late in life (Fig. 3b). The developmental onset of reproduction occurs in the model when follicle count becomes appreciably non-zero and gives the age of 'menarche'. Thus, females ancestrally become fertile early in life with low adult fertility and evolve to become fertile later in life with high adult fertility (Fig. 3b), consistent with empirical analyses[71–76].

Body size ancestrally grows quickly over development and reaches a small size of around 30 kg (blue dots in Fig. 3c) and then evolves so it grows more slowly to a bigger size of around 50 kg (red dots in Fig. 3c), consistent with empirical analyses[71,77]. Body size evolves from a smooth developmental pattern with one growth spurt to a kinked pattern with multiple growth spurts, which are most easily seen as peaks in a weight velocity plot[71,78,79] (Supplementary Fig. 5o, inset). The evolved number and pattern of growth spurts strongly depend on the ancestral genotype (Supplementary Fig. 3o, inset). The evolved age at menarche occurs before the last growth spurt (Fig. 3b,c), in contrast to observation[71] and previous results[37].

Adult skill level evolves expanding from around 2 TB to 4 TB, the units of which arise from the used value of the metabolic cost of memory, which is within an empirically informed range[80] (Fig. 3d).

The evolved developmental growth rates of phenotypic traits are slower than and somewhat different from those observed and those obtained in the previous optimization approach[37], which was already delayed possibly because the developmental Kleiber's law that I use underestimates resting metabolic rate at small body sizes (Fig. C in ref. 36 and Supplementary Fig. 2b in ref. 81). The added developmental delays could be partly due to my use of relatively coarse age bins (0.1 yr) rather than the (nearly) continuous age used previously[37], but halving age bin size (0.05 yr) yields the same outcome (Supplementary Fig. 6). The added developmental delays might also be partly because of slow evolutionary convergence to equilibrium and because the evolved ontogenetic pattern depends on the ancestral genotypic traits (compare the red dots of Fig. 3a,c and Supplementary Fig. 3h,o).

These patterns generate associated expansions in adult brain, body and encephalization quotient (EQ)[49] (Fig. 3e–g). EQ measures

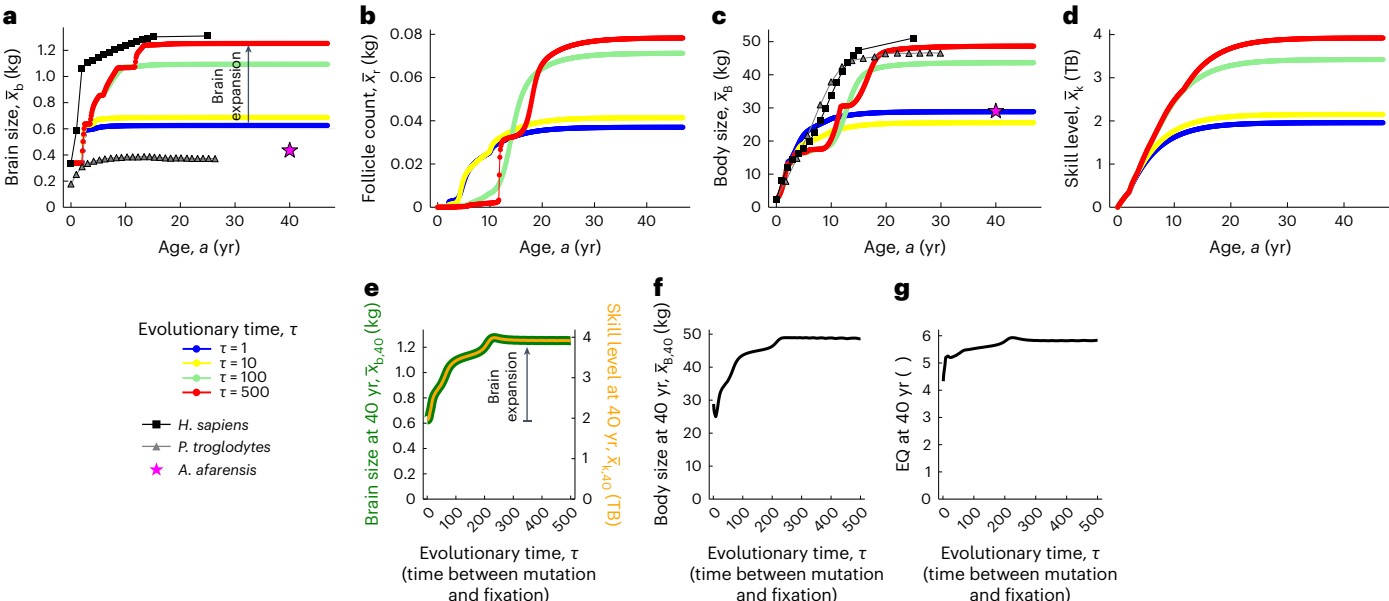

**Fig. 3 | Evo-devo dynamics of hominin brain size. a–d**, Developmental dynamics over age (horizontal axis) and evolutionary dynamics over evolutionary time (differently coloured dots; bottom left label): evo-devo dynamics of brain size (**a**), follicle count (**b**), body size (**c**) and skill level (**d**). **e–g**, Evolutionary dynamics of brain size (green) and skill level (orange) (**e**), body size (**f**) and EQ (**g** non-dimensional) at 40 years of age. In **a** and **c**, the mean observed values in a cross-sectional modern human female sample are shown in black squares (data from Supplementary Table 2 in ref. 39, which fitted data from ref. 69), the mean observed values in cross-sectional *Pan troglodytes* female samples are shown in grey triangles (body size data from Fig. 2 in ref. 107; brain size data from Fig. 6 in

ref. 70), and the mean observed values in *A. afarensis* female samples are shown in pink stars (data from Table 1 in ref. 104). One evolutionary time unit is the time from mutation to fixation. If gene fixation takes 500 generations and 1 generation for females is 23 years[108], then 300 evolutionary time steps are 3.4 million years. The age bin size is 0.1 year. Halving age bin size (0.05 yr) makes the evolutionary dynamics twice as slow, but the system converges to the same evolutionary equilibrium (Supplementary Fig. 6). I take adult phenotypes to be those at 40 years of age as phenotypes have typically plateaued by that age in the model. All plots are for the *sapiens* trajectory of Fig. 2b.

here brain size relative to the expected brain size for a given mammal body size[82]. Adult brain size expands more sharply than adult body size (Fig. 3e,f). Specifically, adult brain size evolves from being ancestrally slightly over four times larger than expected to being about six times larger than expected (Fig. 3g).

The evo-devo dynamics of brain and body sizes that underlie the *afarensis* trajectory in Fig. 2b are shown in Extended Data Fig. 2. The evolved body size under the *afarensis* scenario shows mild indeterminate body growth (red dots in Extended Data Fig. 2c), reminiscent of that in female bonobos (Fig. 6 in ref. 83). Such indeterminate body growth disappears with the plastic change induced by changing to the conditions of the *sapiens* scenario (blue dots in Fig. 3c). The recovered evo-devo dynamics of hominin brain expansion are shown in Supplementary Video 1.

**Analysis of the action of selection**

To understand what causes the obtained brain expansion, I now analyse direct selection and genetic covariation that separate the action of selection and constraint on evolution. Such separation was first formulated for short-term evolution under the assumption of negligible genetic evolution[54,55] and is now available for long-term evolution under non-negligible genetic evolution, clonal reproduction and rare, weak and unbiased mutation[59], as assumed in the present model.

I first analyse the action of selection. In the brain model, fertility is proportional to follicle count, whereas survival is constant as a first approximation. Then, in the brain model, there is always positive direct selection for ever-increasing follicle count, but there is no direct selection for brain size, body size, skill level or anything else (Fig. 4a–d, equation (5) and Extended Data Fig. 1). The fitness landscape has no internal peaks (Fig. 5). Since there is only direct selection for follicle count, the evolutionary dynamics of brain size $\bar{x}_{ba}$ at age $a$ satisfy

$$\frac{d\bar{x}_{ba}}{d\tau} = \iota \sum_{j=1}^{N_a} L_{x_{ba},x_{rj}} \frac{\partial w_j}{\partial x_{rj}}, \tag{1}$$

where $\iota$ is a non-negative scalar measuring mutational input, $L_{x_{ba},x_{rj}}$ ($L$ for legacy) is the mechanistic additive socio-genetic covariance between brain size at age $a$ and follicle count at age $j$, $w_j$ is fitness at age $j$, and $\partial w_j/\partial x_{rj}$ is the direct selection gradient of follicle count at age $j$. Equation (1) shows that brain size evolves in the brain model because brain size is socio-genetically correlated with follicle count (that is, setting the socio-genetic covariation between brain and follicle count to zero in equation (1), so $L_{x_{ba},x_{rj}} = 0$ for all ages $a$ and $j$, yields no brain size evolution).

Brain size and follicle count are socio-genetically correlated in the model because of development. To see this, consider the mechanistic additive socio-genetic cross-covariance matrix of the phenotype, given by

$$\mathbf{L}_x = \text{cov}[\mathbf{b}_x^s, \mathbf{b}_x] = \frac{s\mathbf{x}}{s\mathbf{y}^\top} \mathbf{H}_y \frac{d\mathbf{x}^\top}{d\mathbf{y}}. \tag{2}$$

Here $\mathbf{b}_x$ is the mechanistic breeding value of the phenotype and $\mathbf{b}_x^s$ is the stabilized mechanistic breeding value, which is a generalization of the former and considers the effects of social development. In turn, $\mathbf{H}_y$ is the mutational covariance matrix, $d\mathbf{x}^\top/d\mathbf{y}$ is the matrix of total effects of the genotype on the phenotype, and $s\mathbf{x}/s\mathbf{y}^\top$ is the matrix of stabilized effects of the genotype on the phenotype, where stabilized effects are the total effects after social development has stabilized in the population. Whereas $\mathbf{H}_y$ depends on genotypic traits but not development, both $d\mathbf{x}^\top/d\mathbf{y}$ and $s\mathbf{x}/s\mathbf{y}^\top$ depend on development. In the model, there is no mutational covariation (that is, $\mathbf{H}_y$ is diagonal), so $\mathbf{H}_y$ does

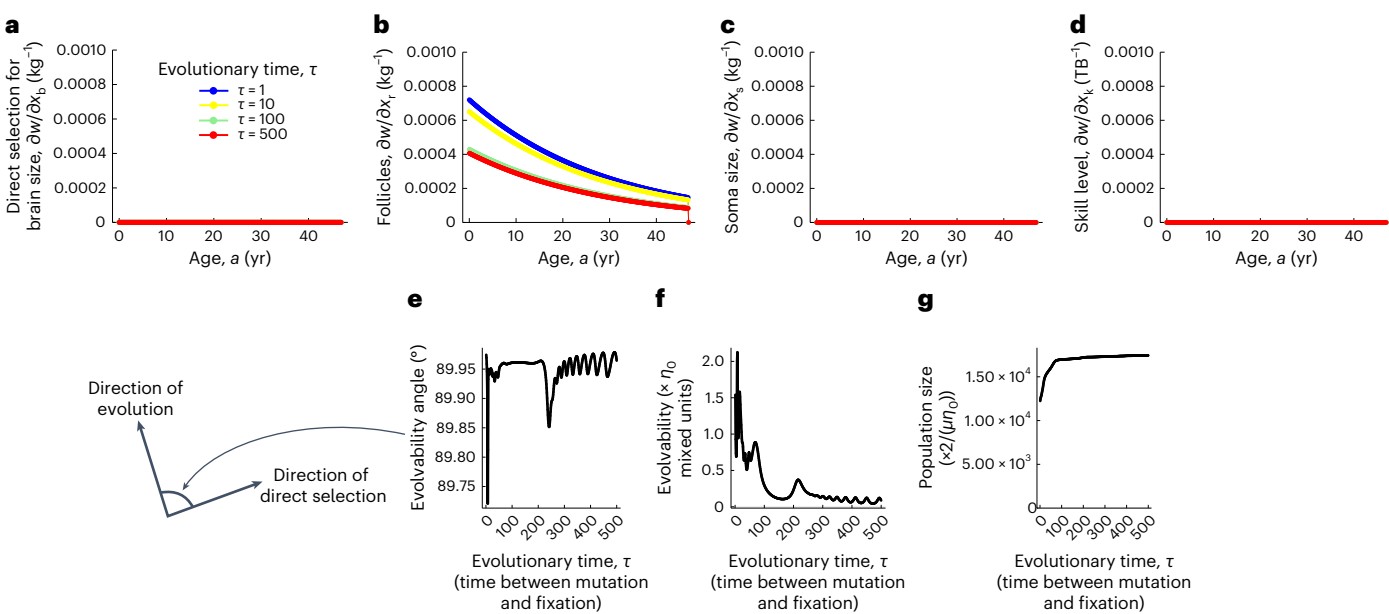

**Fig. 4 | The action of selection. a–d**, There is no direct selection for brain size (**a**), somatic tissue size (**c**), nor skill level (**d**). There is direct selection only for follicle count, and such selection decreases with age (**b**). **e**, The angle between the direction of evolution and direct selection, both of the geno–phenotype (that is, genotype and phenotype), is nearly 90 degrees over evolutionary time. **f**, Evolvability is small and decreases over evolutionary time. Evolvability equal to 0 here means no evolution despite selection (Supplementary Section 7 and equation (1) in ref. 84). **g**, Population size increases over evolutionary time (plot

of $\frac{1}{2}\mu\bar{n}^{*}\eta_{0}$, so the indicated multiplication yields population size). Mutation rate $\mu$ and parameter $\eta_{0}$ can take any value satisfying $0 < \mu \ll 1$ and $0 < \eta_{0} \ll 1/(N_{g}N_{a})$, where the number of genotypic traits is $N_{g} = 3$ and the number of age bins is $N_{a} = 47\,\text{yr}/0.1\,\text{yr} = 470$. If $\mu = 0.01$ and $\eta_{0} = 1/(3 \times 47\,\text{yr}/0.1\,\text{yr})$, then a population size of $1{,}000 \times 2/(\mu\eta_{0})$ is 2.82 billion individuals (which is unrealistically large owing to the assumption of marginally small mutational variance to facilitate analysis). All plots are for the *sapiens* trajectory of Fig. 2b.

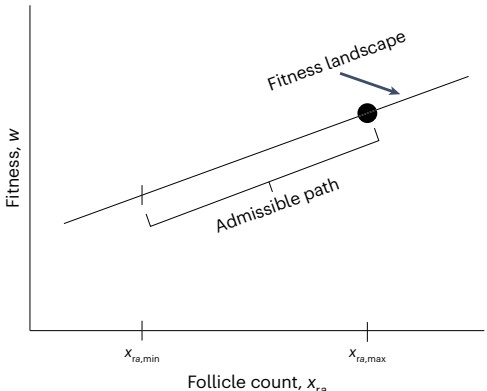

**Fig. 5 | Illustration of the fitness landscape in the brain model.** The fitness landscape $w$ is a linear function (equation (5)) of the follicle count $x_{ra} = g_{r,a-1}(\mathbf{x}_{a-1}, \mathbf{y}_{a-1}, \bar{x}_{k,a-1})$, which is a recurrence over age. The slope of the fitness landscape with respect to $x_{ra}$ is positive and decreases with age $a$ (Fig. 4b). Evaluating the recurrence at all possible genotypic trait values $\mathbf{y}_{j} \in \mathbb{R}^{N_{g}}$ for all ages $j < a$ gives values $x_{ra,\min}$ and $x_{ra,\max}$ that depend on development $g_{rj}$ for all ages $j < a$, the various parameters influencing it, and the developmentally initial conditions. The admissible follicle count ranges from $x_{ra,\min}$ to $x_{ra,\max}$. The admissible path on the landscape is given by the admissible follicle count. As there are no absolute mutational constraints, evolution converges to the peak of the admissible path[60] (dot), where total genotypic selection vanishes, $d w/d\mathbf{y} = \mathbf{0}$ (Extended Data Fig. 3).

not generate socio-genetic covariation between brain size and follicle count. Hence, such socio-genetic covariation can only arise from the total and stabilized effects of the genotype on the phenotype, which arise from development.

Therefore, the various evolutionary outcomes matching the brain and body sizes of seven hominin species[37] (Fig. 1) arise in this model exclusively owing to change in developmental constraints and not from change in direct selection on brain size or cognitive abilities. In the model, EETBs and the shape of EEE only directly affect the developmental map ($\mathbf{g}_{a}$) but not fitness, so varying EETBs and the shape of EEE does not affect the direction of direct selection but only its magnitude (Supplementary equation (38)). Moreover, from the equation that describes the long-term evolutionary dynamics (equation (7)), it follows that varying EETBs and the shape of EEE only affects evolutionary outcomes (that is, path peaks; Fig. 5) by affecting the mechanistic socio-genetic covariation $\mathbf{L}_{z}$ (Supplementary equation (29)). That socio-genetic covariation determines evolutionary outcomes despite no internal fitness landscape peaks is possible because there is socio-genetic covariation only along the admissible path where the developmental constraint is met (so $\mathbf{L}_{z}$ is singular[59]) and consequently evolutionary outcomes occur at path peaks rather than landscape peaks[60] (Fig. 5). That is, the various evolutionary outcomes matching the brain and body sizes of seven hominin species[37] (Fig. 1) are exclusively due to change in mechanistic socio-genetic covariation described by the $\mathbf{L}_{z}$ matrix, by changing the position of path peaks on the peak-invariant fitness landscape. Hence, ecology and possibly culture cause hominin brain expansion in the model by affecting developmental and consequently socio-genetic constraints rather than direct selection. In addition, brain metabolic costs directly affect the developmental map ($\mathbf{g}_{a}$) and so affect mechanistic socio-genetic covariation ($\mathbf{L}_{z}$) but do not directly affect fitness ($w$) and so do not constitute direct fitness costs (Supplementary equations (2) and (8–10), and equation (5)). Yet, in the model, brain metabolic costs often constitute total fitness costs and, occasionally, total fitness benefits (Methods, Supplementary Fig. 7 and Supplementary Section 8).

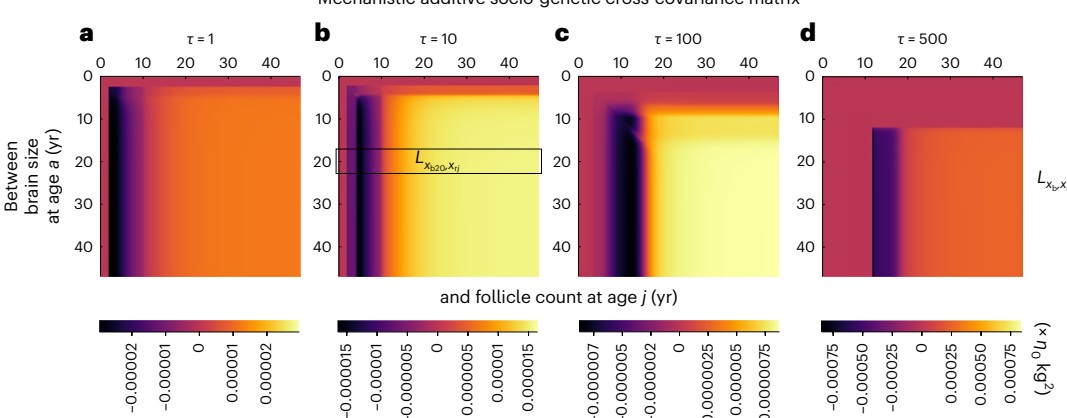

Mechanistic additive socio-genetic cross-covariance matrix

**Fig. 6 | The action of constraint on brain expansion. a–d,** Mechanistic socio-genetic cross-covariance matrix between brain size and follicle count at evolutionary time $\tau = 1$ (**a**), $\tau = 10$ (**b**), $\tau = 100$ (**c**) and $\tau = 500$ (**d**) for the *sapiens* trajectory of Fig. 2b. For instance, in **b**, the highlighted box gives the socio-genetic covariance between brain size at 20 years of age and follicle count at each of the ages at the top horizontal axis. Thus, at evolutionary time $\tau = 10$, socio-genetic covariation between brain size at 20 years of age and follicle count at 6 years of age is negative (bottom bar legend) but between brain size at 20 years of age and follicle count at 30 years of age is positive. The positive socio-genetic covariation between brain size and follicle count (for example, yellow areas in **b** and **c**) causes brain expansion. Bar legends have different limits so that patterns are visible (bar legend limits are $\{-l, l\}$, where $l = \max(|L_{x_{ba}, x_{\tau j}}|)$ over $a$ and $j$ for each $\tau$).

Evolution is almost orthogonal to direct selection throughout hominin brain expansion in the model (Fig. 4e). Evolvability[84], measuring the extent to which evolution proceeds in the direction of direct selection, is ancestrally very small and decreases towards zero as evolution proceeds (Fig. 4f), but increases with the plastic change in the phenotype when shifting from the *afarensis* scenario to the *sapiens* scenario (Supplementary Fig. 8m). Thus, evolution stops because there is no longer socio-genetic variation in the direction of direct selection. Despite the absence of direct selection on brain size or skill level in the model, there is total selection on the various traits, but total selection confounds the action of selection and constraint (Extended Data Fig. 3 and Methods). The population size expands as the brain expands (Fig. 4g), although it decreases when shifting from the *afarensis* scenario to the *sapiens* scenario owing to the plastic change in phenotype (Supplementary Fig. 8n).

**Analysis of the action of constraint**

To gain further insight into what causes the recovered brain expansion, I now analyse the action of constraint. Since there is only direct selection for follicle count, the equation describing long-term evolution (equation (7)) entails that whether or not a trait evolves in the model is dictated by whether or not there is mechanistic socio-genetic covariation between the trait and follicle count (for example, equation (1)).

Examination of such covariation shows that brain expansion in the model is caused by positive socio-genetic covariation between brain size and developmentally late follicle count. The mechanistic socio-genetic covariation of brain size with follicle count, and how such covariation evolves, is shown in Fig. 6. Ancestrally, socio-genetic covariation between brain size and developmentally early follicle count is negative (black area in Fig. 6a) but between brain size and developmentally late follicle count is slightly positive (orange area in Fig. 6a). This positive covariation is what causes brain expansion. This pattern of socio-genetic covariation is maintained as brain expansion proceeds, but developmentally early brain size becomes less socio-genetically covariant with follicle count and so stops evolving, whereas developmentally later brain size becomes socio-genetically covariant with increasingly developmentally later follicle count. The magnitude of covariation also evolves (Fig. 6a–d).

Hence, direct selection on developmentally late follicle count provides a force for follicle count increase, and socio-genetic covariation between brain size and developmentally late follicle count diverts this force and causes brain expansion. This occurs even though the force of selection is weaker at advanced ages[85] (that is, slopes are negative in Fig. 4b), which is compensated by developmentally increasing socio-genetic covariation with follicle count. Such increasing covariation can arise because of developmental propagation of phenotypic effects of mutations[60]. Therefore, ecology and culture cause brain expansion in the model by generating positive socio-genetic covariation over development between brain size and developmentally late follicle count.

The socio-genetic covariation between body size and follicle count, and between skill level and follicle count, follow a similar pattern (Extended Data Fig. 4a–h). Hence, the evolutionary expansion in body size and skill level in the model are also caused by their positive socio-genetic covariation with developmentally late follicle count.

The evolution of follicle count is governed by a different pattern of socio-genetic covariation. Developmentally early follicle count evolves smaller values because of negative socio-genetic covariation with developmentally late follicle count (Extended Data Fig. 4i–l). In turn, developmentally late follicle count evolves higher values because of positive socio-genetic covariation with developmentally late follicle count (Extended Data Fig. 4i–l). Positive socio-genetic covariance between follicle count of different ages is clustered at the ages where follicle count developmentally increases most sharply (Fig. 3b). This cluster of positive socio-genetic covariation evolves to later ages (Extended Data Fig. 4j–l), corresponding to the evolved ages of peak developmental growth in follicle count (Fig. 3b). This cluster of positive socio-genetic covariation has little effect on follicle count evolution as follicle count around the evolving age of menarche mostly decreases over evolution, so such covariation is mostly compensated by the negative socio-genetic covariation with developmentally later follicle count. Socio-genetic covariation between other phenotypes exists (Supplementary Figs. 9 and 10) but has no evolutionary effect as only that with follicle count does. Several of the above patterns of socio-genetic covariation emerge during the *afarensis* trajectory (Supplementary Fig. 11).

## Discussion

I modelled the evo-devo dynamics of hominin brain expansion, recovering major patterns of human development and evolution. I showed that hominin brain expansion occurs in this model because brain size is socio-genetically correlated with developmentally late follicle count and there is only direct selection for follicle count. In other words, mutant alleles coding for increased allocation to brain growth can only increase in frequency in the model by being socio-genetically correlated with developmentally late follicle count that is selected for, rather than owing to direct selection for brain size. This socio-genetic correlation is generated over development by a moderately challenging ecology and possibly cumulative culture. This covariation yields an admissible evolutionary path on the fitness landscape (Fig. 5), a path along which the brain expands because of developmental constraints, as without them there is no direct selection for or against brain expansion. Thus, in this model, hominin brain expansion is caused by unremarkable selection and particular developmental constraints involving a moderately challenging ecology and possibly cumulative culture. This constraint-caused brain expansion occurs despite it generating a brain–body allometry of 1.03 and a duplication of EQ (Extended Data Fig. 2g and Fig. 3g). While cognitive ability in the form of skill level is not directly under selection in the model, the model can be modified to incorporate such widely considered scenario. Yet, as found above, direct selection for cognitive ability is not necessary to recreate hominin brain expansion and multiple aspects of human development and evolution, whereas certain developmental constraints with unexceptional direct selection are sufficient, at least for the parameter values analysed. Change in development without changes in direct selection can thus yield diverse evolutionary outcomes, including the brain and body sizes of seven hominins, rather than only evolutionarily transient effects.

These results show that developmental constraints can play major evolutionary roles by causing hominin brain expansion in this in silico replica. Developmental constraints are traditionally seen as preventing evolutionary change[12,52,86,87], effectively without the ability to generate evolutionary change that is not already favoured by selection. Less prevalent views have highlighted the potential relevance of developmental constraints in evolution[51] and human brain evolution (for example, page 87 in ref. 88). The findings here show that while constraints do prevent evolutionary change in some directions, constraints can be 'generative'[89] in the sense that they can divert evolutionary change in a direction that causes brain expansion, such that without those constraints, brain expansion is not favoured by selection and does not evolve.

The results above contrast with a previous study, finding that direct selection on brain size must have driven hominin brain expansion[58]. Such a study assumed that direct selection for adult brain and/or body sizes generated the observed data. My approach differs by simulating the data and determining what selection occurs in the simulation. This finds that direct selection for adult brain or body sizes is unnecessary to replicate major patterns of hominin brain expansion.

My approach illustrates why the human brain size could have evolved, but it has not established why it did. Yet, this approach can be built upon to pursue that goal. There is scope for refinement of the model, for improved parameter estimates, for relaxing assumptions and for other models to improve predictions as those obtained are near but do not exactly match observation, particularly in the ontogenetic patterns. With more models, rapidly advancing techniques of simulation-based inference may be used for model selection, parameter estimation and uncertainty quantification[31]. These techniques have been instrumental in multiple fields such as in the discovery of the Higgs boson[31] or in establishing that humans are causing climate change, and my results suggest that simulation-based inference with the brain model is now within reach. Indeed, simulation-based inference with the brain model was previously impractical with the dynamic optimization approach, as a single run took approximately 3 days[37], the runs are not easy to parallelize as suitable initial guesses are needed for the genotypic traits, and simulation-based inference needs hundreds of thousands of runs. This meant that simulation-based inference would have taken about 800 years. By contrast, a run here took approximately 4 minutes, indicating that simulation-based inference with the evo-devo dynamics approach could take months. This computational speed suggests that simulation-based inference[31] of human brain size evolution may now be feasible.

## Methods

### Model overview

The evo-devo dynamics framework that I use[59] is based on standard adaptive dynamics assumptions[90,91]. The framework considers a resident, well-mixed, finite population with deterministic population dynamics where individuals can be of different ages, reproduction is clonal, and mutation is rare (mutants arise after previous mutants have fixed) and weak (mutant genotypes are marginally different from the resident genotype). Under these assumptions, population dynamics occur in a fast ecological timescale and evolutionary dynamics occur in a slow evolutionary timescale. Individuals have genotypic traits, collectively called the genotype, that are under direct genetic control. As mutation is weak, there is vanishingly small variation in genotypic traits (marginally small mutational variance). Also, individuals have phenotypic traits, collectively called the phenotype, that are developed, that is, constructed over life. A function $\mathbf{g}_a$, called the developmental map, describes how the phenotype is constructed over life and gives the developmental constraint. The developmental map can be non-linear, evolve, change over development and take any differentiable form with respect to its arguments, but the phenotype at the initial age (here newborns) is constant and does not evolve, as is standard in life history theory. Mutant individuals of age $a$ have fertility $f_a$ (rate of offspring production) and survive to the next age with probability $p_a$. The evo-devo dynamics framework provides equations describing the evolutionary dynamics of genotypic and phenotypic traits in gradient form, thus describing long-term genotypic and phenotypic evolution as the climbing of a fitness landscape while guaranteeing that the developmental constraint is met at all times.

The brain model[36,37] provides a specific developmental map $\mathbf{g}_a$, fertility $f_a$ and survival $p_a$, which can be fed into the evo-devo dynamics framework to model the evolutionary dynamics of the developed traits studied. More specifically, the brain model considers a female population, where each individual at each age has three tissue types—brain, reproductive and remaining somatic tissues—and a skill level. Reproductive tissue is defined as referring to preovulatory ovarian follicles, so that reproductive tissue is not involved in offspring maintenance, which allows for writing fertility as being proportional to follicle count (in mass units), in accordance with observation[92]. As a first approximation, the brain model lets the survival probability at each age be constant. At each age, each individual has an energy budget per unit time, her resting metabolic rate $B_{rest}$, that she uses to grow and maintain her tissues. The part of this energy budget used in growing her tissues is her growth metabolic rate $B_{syn}$. A fraction of the energy consumed by the preovulatory follicles is for producing offspring, whereas a fraction of the energy consumed by the brain is for gaining (learning) and maintaining (memory) skills. Each individual's skill level emerges from this energy bookkeeping rather than being assumed as given by brain size. Somatic tissue does not have a specific function but it contributes to body size, thus affecting the energy budget because of Kleiber's law[93], which relates resting metabolic rate to body size by a power law. Genes control the individual's energy allocation effort into producing brain tissue, preovulatory follicles and somatic tissue at each age. The causal dependencies in the brain model are described in Extended Data Fig. 1, which uses the insights from the evo-devo dynamics framework, in particular, the separation of direct and total effects on fitness in the model.

I write the brain model with the notation of the evo-devo dynamics framework as follows. The model considers four phenotypic traits (that is, $N_p = 4$): brain mass, follicle count (in mass units), somatic tissue mass and skill level at each age. For a mutant individual, the brain size at age $a \in \{1, \dots, N_a\}$ is $x_{ba}$ (in kilograms), the follicle count at age $a$ is $x_{ra}$ (in kilograms), the size of the remaining somatic tissue at age $a$ is $x_{sa}$ (in kilograms), and the skill level at age $a$ is $x_{ka}$ (in terabytes (TB)). The units of phenotypic traits (kg and TB) arise from the units of the parameters measuring the unit-specific metabolic costs of maintenance and growth of the respective trait. The vector $\mathbf{x}_a = (x_{ba}, x_{ra}, x_{sa}, x_{ka})^\top$ is the mutant phenotype at age $a$. In addition, the model considers three genotypic traits (that is, $N_g = 3$): the effort to produce brain tissue, preovulatory follicles and somatic tissue at each age. For a mutant individual, the effort at age $a$ to produce brain tissue is $y_{ba}$, follicles is $y_{ra}$, and somatic tissue is $y_{sa}$. These growth efforts are dimensionless and can be positive or negative, so they can be seen as measured as the difference from a baseline growth effort. The model assumption that the growth efforts $y_{ia}$ are genotypic traits that can vary with age can be understood as each $y_{ia}$ being determined by the individual's genotype at a separate locus, so there are $N_g N_a = 3 \times 470 = 1{,}410$ loci. The vector $\mathbf{y}_a = (y_{ba}, y_{ra}, y_{sa})^\top$ is the mutant growth effort at age $a$, which describes the mutant genotypic traits at that age. The growth efforts generate the fraction $q_{ia}(\mathbf{y}_a)$ of the growth metabolic rate $B_{syn}$ allocated to growth of tissue $i \in \{b, r, s\}$ at age $a$ ($q_{ia}$ corresponds to the control variables $u$ in refs. 36,37; I consider $y$'s rather than $q$'s as the genotypic traits as the $y$'s do not need to be between zero and one nor add up to one, so numerical solution is simpler). To describe the evolutionary dynamics of the phenotype as the climbing of a fitness landscape, the evo-devo dynamics framework defines the mutant geno–phenotype at age $a$ as the vector $\mathbf{z}_a = (\mathbf{x}_a; \mathbf{y}_a)$ (the semicolon indicates a line break). The mutant phenotype across ages is $\mathbf{x} = (\mathbf{x}_1; \dots; \mathbf{x}_{N_a})$ and similarly for the other variables. While $\mathbf{x}_a$ is a mutant's phenotype across traits at age $a$, I denote the mutant's $i$th phenotype across ages as $\mathbf{x}_{i\bullet} = (x_{i1}, \dots, x_{iN_a})^\top$ for $i \in \{b, r, s, k\}$ and similarly for the other variables. The resident traits are analogously denoted with an overbar (for example, $\bar{\mathbf{x}}$).

The brain model describes development by providing equations describing the developmental dynamics of the phenotype. That is, the mutant phenotype at age $a + 1$ is given by the developmental constraint

$$\mathbf{x}_{a+1} = \mathbf{g}_a(\mathbf{x}_a, \mathbf{y}_a, \bar{x}_{ka}). \tag{3}$$

The equations for the developmental map $\mathbf{g}_a$ are given in Supplementary Section 1.1 and were previously derived from mechanistic considerations of energy conservation following the reasoning of West et al.'s metabolic model of ontogenetic growth[38] and phenomenological considerations of how skill relates to energy extraction[36,37]. The developmental map of the brain model depends on the skill level of social partners of the same age (that is, peers), $\bar{x}_{ka}$, because of social challenges of energy extraction (where $P_1 < 1$), so I say that development is social. When individuals face only ecological challenges (that is, $P_1 = 1$), development is not social.

The evo-devo dynamics are described by the developmental dynamics of the phenotypic traits given by equation (3) and by the evolutionary dynamics of the genotypic traits given by the canonical equation of adaptive dynamics[90]

$$\frac{\Delta \bar{\mathbf{y}}}{\Delta \tau} = \iota \mathbf{H}_y \frac{dw}{d\mathbf{y}}, \tag{4}$$

where $\tau$ is the evolutionary time, $\iota$ is a non-negative scalar measuring mutational input and is proportional to the mutation rate and carrying capacity, and $\mathbf{H}_y = \text{cov}[\mathbf{y}, \mathbf{y}]$ is the mutational covariance matrix ($\mathbf{H}$ for heredity; derivatives are evaluated at resident trait values throughout and I use matrix calculus notation[94] as defined in Supplementary

equation (1)). Owing to age structure, a mutant's relative fitness is $w = \sum_{a=1}^{N_a} w_a = \frac{1}{T} \sum_{a=1}^{N_a} (\phi_a f_a + \pi_a p_a)$, where $f_a$ and $p_a$ are a mutant's fertility and survival probability at age $a$, $T$ is the generation time, and $\phi_a$ and $\pi_a$ are the forces[85,95,96] of selection on fertility and survival at that age ($T$, $\phi_a$ and $\pi_a$ are functions of the resident but not mutant trait values). After substitution and simplification, a mutant's relative fitness reduces to

$$w = \frac{1}{\sum_{a=1}^{N_a} ap^{a-1} \bar{x}_{ra}} \sum_{j=1}^{N_a} \left( p^{j-1} x_{rj} + \sum_{k=j+1}^{N_a} p^{k-1} \bar{x}_{rk} \right), \tag{5}$$

where $p$ is the constant probability of surviving from one age to the next. Hence, this fitness function depends directly on the mutant's follicle count but only indirectly on metabolic costs via the developmental constraint (that is, after substituting $x_{rj}$ for the corresponding entry of equation (3)).

Equation (4) depends on the total selection gradient of genotypic traits $dw/d\mathbf{y}$, which measures total genotypic selection. While Lande's[54] selection gradient measures direct selection without considering developmental constraints by using partial derivatives ($\partial$), total selection gradients measure selection considering developmental constraints (equation (3)) by using total derivatives (d). The total selection gradient of genotypic traits for the brain model is

$$\frac{dw}{d\mathbf{y}} = \frac{\partial \mathbf{x}^\top}{\partial \mathbf{y}} \frac{dw}{d\mathbf{x}} = \frac{d\mathbf{x}^\top}{d\mathbf{y}} \frac{\partial w}{\partial \mathbf{x}} \tag{6}$$

(from the first and last equalities in Layer 4, Supplementary equation S1 in ref. 59). Equation (6) shows that total genotypic selection can be written in terms of either total phenotypic selection ($dw/d\mathbf{x}$) or direct phenotypic selection ($\partial w/\partial \mathbf{x}$). Equation (4) entails that total genotypic selection vanishes at evolutionary equilibria if there are no absolute mutational constraints (that is, if $\iota > 0$ and $\mathbf{H}_y$ is non-singular). Moreover, since in the brain model there are more phenotypic traits than genotypic traits ($N_p > N_g$), the matrices $\partial \mathbf{x}^\top/\partial \mathbf{y}$ and $d\mathbf{x}^\top/d\mathbf{y}$ have fewer rows than columns and so are singular; hence, setting equation (6) to zero implies that evolutionary equilibria can occur with persistent direct and total phenotypic selection in the brain model.

While I use equations (3) and (4) to compute the evo-devo dynamics, those equations do not describe phenotypic evolution as the climbing of an adaptive topography. To analyse phenotypic evolution as the climbing of an adaptive topography, I use the following. The evo-devo dynamics framework[59] shows that long-term phenotypic evolution can be understood as the climbing of a fitness landscape by simultaneously following genotypic and phenotypic evolution, which for the brain model is given by

$$\frac{d\bar{\mathbf{z}}}{d\tau} = \iota \mathbf{L}_z \frac{\partial w}{\partial \mathbf{z}}, \tag{7}$$

since $\mathbf{z} = (\mathbf{x}; \mathbf{y})$ includes the phenotype $\mathbf{x}$ and genotypic traits $\mathbf{y}$. The vector $\partial w/\partial \mathbf{z}$ is the direct selection gradient of the geno–phenotype (as in Lande's[54] selection gradient of the phenotype). The matrix $\mathbf{L}_z$ is the mechanistic additive socio-genetic cross-covariance matrix of the geno–phenotype, for which the evo-devo dynamics framework provides formulas that guarantee that the developmental constraint (3) is met at all times. The matrix $\mathbf{L}_z$ is asymmetric owing to social development; if individuals face only ecological challenges, development is not social and $\mathbf{L}_z$ reduces to $\mathbf{H}_z$, the mechanistic additive genetic covariance matrix of the geno–phenotype, which is symmetric ($\mathbf{H}_x$ is a mechanistic version of Lande's[54] $\mathbf{G}$ matrix: whereas $\mathbf{G}$ is defined in terms of the linear regression of phenotype on genotype, $\mathbf{H}_x$ involves total derivatives describing the total effect of genotype on phenotype; hence, $\mathbf{H}_x$ and $\mathbf{G}$ have different properties, including that mechanistic heritability can be greater than one). The matrix $\mathbf{L}_z$ is always singular because it

considers both the phenotype and genotypic traits, so selection and development jointly define the evolutionary outcomes even with a single fitness peak[60]. Equation (7) and the formulas for $\mathbf{L}_z$ entail that evolution proceeds as the climbing of the fitness landscape in geno-phenotype space, where the developmental constraint (3) provides the admissible evolutionary path, such that evolutionary outcomes occur at path peaks rather than landscape peaks if there are no absolute mutational constraints[60].

I implement the developmental map of the brain model into the evo-devo dynamics framework to study the evolutionary dynamics of the resident phenotype $\bar{\mathbf{x}}$, including the resident brain size $\bar{x}_{b}$..

### Seven hominin scenarios

It was previously found[37] that, at evolutionary equilibrium, the brain model recovers the evolution of the adult brain and body sizes of six *Homo* species and less accurately of *A. afarensis*. The parameter values yielding these seven outcomes are described in Fig. 1. I call each such parameter combination a scenario. The *sapiens*, *neanderthalensis* and *heidelbergensis* scenarios use weakly diminishing returns of learning and submultiplicative cooperation: specifically, these scenarios use exponential competence with parameter values given in Regime 1 of Supplementary Table 1 (Supplementary equation (5)). I call ecological scenario that with such weakly diminishing returns of learning and submultiplicative cooperation but setting the proportion of ecological challenges to one ($P_1 = 1$), which was previously[36] found to yield the evolution of brain and body sizes of the Neanderthal scale at evolutionary equilibrium. The *erectus*, *ergaster* and *habilis* scenarios use strongly diminishing returns of learning and additive cooperation: specifically, these scenarios use power competence with parameter values given in Regime 2 of Supplementary Table 1 and with additive cooperation (Supplementary equation (5)). The *afarensis* scenario uses strongly diminishing returns of learning and submultiplicative cooperation, that is, power competence with parameter values given in Regime 2 of Supplementary Table 1 (Supplementary equation (5)). In the main text, I primarily describe results under the *sapiens* scenario. In Supplementary Information, I also give analogous results under the *afarensis* (Supplementary Figs. 4, 8 and 11) and ecological (Supplementary Fig. 2) scenarios.

### Ancestral genotypic traits

To solve the evo-devo dynamics, one must specify the ancestral resident genotypic traits giving the resident growth efforts $\bar{\mathbf{y}}$ at the initial evolutionary time. I explored nine sets of ancestral genotypes described in Supplementary Section 4 and labelled as naive, somewhatNaive, ecoSols, highlySpecified, afarensisFromHighlySpecified, afarensisFromSomewhatNaive, somewhatNaive2, afarensisFromNaive2 and afarensisFromEcoSols.

I find that the outcome depends on the ancestral genotype. For instance, in the *sapiens* scenario, at least two drastically different evolutionary outcomes are possible by changing only the ancestral genotype (that is, there is bistability in brain size evolution), so there are at least two path peaks on the fitness landscape as follows. Using somewhatNaive ancestral growth efforts in the *sapiens* scenario yields an evolutionary outcome with no brain, where residents have a somewhat semelparous life history reproducing for a short period early in life followed by body shrinkage (Supplementary Fig. 1). By contrast, using afarensisFromNaive2 ancestral growth efforts in the *sapiens* scenario yields adult brain and body sizes of the *H. sapiens* scale (Fig. 3). This bistability does not arise under the ecological scenario, which yields brain expansion under somewhatNaive ancestral growth efforts (Supplementary Fig. 2). Thus, for the *sapiens* scenario to yield brain and body sizes of the *H. sapiens* scale, it seems to require an ancestral genotype that can develop large bodies under cooperative and between-group competitive social challenges (blue dots in Fig. 3c and Supplementary Fig. 3o), whereas ancestrally large bodies are not

needed for brain expansion under purely ecological challenges (blue dots in Supplementary Fig. 2o). In the main text, I present the results for the *sapiens* scenario with the afarensisFromNaive2 ancestral genotype.

### The action of total selection

Total selection is measured by total selection gradients that quantify the total effect of a trait on fitness considering the developmental constraints and so how traits affect each other over development[59,97]. Thus, in contrast to direct selection, total selection confounds the action of selection and constraint. Since I assume there are no absolute mutational constraints (that is, $\mathbf{H}_y$ is non-singular), evolutionary outcomes occur at path peaks in the fitness landscape where total genotypic selection vanishes ($dw/d\mathbf{y} = \mathbf{0}$), which are not necessarily fitness landscape peaks where direct selection vanishes ($\partial w/\partial \mathbf{z} \neq \mathbf{0}$).

The following patterns of total selection occur during the *sapiens* trajectory of Fig. 2b. Total selection ancestrally favours increased brain size throughout life (blue circles in Extended Data Fig. 3a). As evolution advances, total selection for brain size decreases and becomes negative early in life, possibly owing to my assumption that the brain size of a newborn is fixed and cannot evolve. A similar pattern results for total selection on follicle count (Extended Data Fig. 3b). Somatic tissue is ancestrally totally selected for early in life and against later in life, eventually becoming totally selected for throughout life (Extended Data Fig. 3c). Total selection for skill level ancestrally fluctuates but decreases across life, decreasing as evolution proceeds but remaining positive throughout life (Extended Data Fig. 3d). Thus, total selection still favours evolutionary change in the phenotype at evolutionary equilibrium, but change is no longer possible (red dots in Extended Data Fig. 3a–d are at non-zero values). This means that evolution does not and cannot reach the favoured total level of phenotypic change in the model.

Although evolution does not reach the favoured total level of phenotypic change in the model, it does reach the favoured total level of genotypic change because of the assumption of no absolute mutational constraints. Total selection for the genotypic trait of brain growth effort is ancestrally positive early in life and evolves towards zero (Extended Data Fig. 3e). Total genotypic selection for follicle production is also ancestrally positive early in life, transiently evolves to negative and eventually approaches zero (Extended Data Fig. 3f). Total genotypic selection for somatic growth effort is ancestrally negative early in life and evolves towards zero (Extended Data Fig. 3g). The evolved lack of total genotypic selection means that evolution approaches the favoured total level of genotypic change. This also means that evolution stops at a path peak on the fitness landscape (Fig. 5).

The occurrence of total selection for brain size or skill level may suggest that this total selection causes brain expansion in the model, but in the recovered brain expansion, total selection can change the evolved brain size only owing to change in the developmental constraints. This is because total selection equals the product of direct selection and total developmental bias (equation (6) and Supplementary equation (34)), and in the model changing EETBs or the shape of EEE does not affect the direction of direct selection but only the direction of total developmental bias by affecting the developmental constraints. Thus, varying EETBs or the shape of EEE affects total selection in the evolved brain and body sizes only because the developmental constraints are changed rather than direct selection.

### Total fitness effects of metabolic costs

While brain metabolic costs do not entail direct fitness costs in the model (that is, $\partial w/\partial B_b = 0$), they may entail total fitness costs (that is, $dw/dB_b \neq 0$), and these can be computed using formulas from the evo-devo dynamics framework (Supplementary Section 8). Using these formulas shows that metabolic costs of maintenance may be total fitness costs at some ages but benefits at some other ages over the *sapiens* trajectory (Supplementary Fig. 7). Consequently, the metabolic cost

of brain maintenance is a total fitness benefit at evolutionary time 1 ($dw/dB_b = \sum_{a=1}^{N_a} dw/dB_{ba} = 2.1 \times 10^{-6}$ kg yr$^{-1}$ MJ$^{-1}$) and a total fitness cost at evolutionary times 10, 100 and 500 ($-1.5 \times 10^{-5}$ kg yr$^{-1}$ MJ$^{-1}$, $-2 \times 10^{-5}$ kg yr$^{-1}$ MJ$^{-1}$ and $-1.7 \times 10^{-5}$ kg yr$^{-1}$ MJ$^{-1}$, respectively). Moreover, among tissues, the metabolic cost of somatic maintenance has some of the most substantial total fitness effects, even though it is the smallest metabolic cost of maintenance (Supplementary Fig. 7i–l), perhaps owing to the large size of somatic tissue. Total fitness costs also confound the action of selection and constraint as they depend on development rather than only on selection. That is, total fitness costs share components with genetic covariation.

### Reporting summary

Further information on research design is available in the Nature Portfolio Reporting Summary linked to this article.

### Data availability

No data were collected in this study. All data used were previously published in references provided in the main text or Supplementary Information.

### Code availability

All codes are available in Supplementary Information and have been deposited at https://doi.org/10.5281/zenodo.10887414 (ref. 62).

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

## Acknowledgements

I thank A. Gardner, K. N. Lala (formerly Laland), R. Patchett and C. R. Turner for comments on previous versions of the paper, A. Gardner for funding, and S. D. Healy, P. Gunz and C. Rutz for discussion. A. Gardner suggested to randomly sample genotypic traits to evaluate the resulting brain–body allometry as in Fig. 2a. This work was funded by a European Research Council Consolidator Grant to A. Gardner (grant no. 771387). The funders had no role in study design, data collection and analysis, decision to publish or preparation of the paper.

## Competing interests

The author declares no competing interests.

## Additional information

**Extended data** is available for this paper at https://doi.org/10.1038/s41562-024-01887-8.

**Correspondence and requests for materials** should be addressed to Mauricio González-Forero.

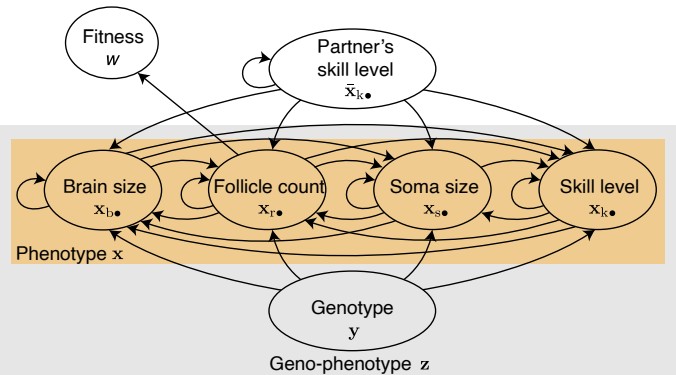

**Extended Data Fig. 1 | Causal diagram of the brain model analysed under the evo-devo dynamics framework.** The evo-devo dynamics framework clarifies how to separate the direct and total effects of traits on fitness in the model. Variables have age-specific values. The phenotype comprises brain size, follicle count, somatic tissue size, and skill level, all constructed by a developmental process. Each arrow indicates the direct effect of a variable on another one. The total effect of a variable on another one is that across all the arrows directly or indirectly connecting the former to the latter. A mutant's genotypic traits at a given age directly affect brain size, follicle count, somatic tissue size, and skill level at the immediately subsequent age (with the slope quantifying developmental bias from genotype). A mutant's phenotypic traits at a given age affect themselves at the immediately subsequent age (quantifying developmental bias from the same phenotypic trait), thus the direct feedback loop from phenotypic traits to themselves. A mutant's phenotypic traits at a given age also directly affect each other at the next age (quantifying developmental bias from immediately previous phenotypes). A mutant's follicle count is the only trait directly affecting fitness (direct selection on follicle count). The social partner's skill level at a given age directly affects its own development at an immediately subsequent age (quantifying developmental bias from the same phenotypic trait), thus the direct feedback loop. The social partner's skill level at a given age also directly affects all the mutant's phenotypic traits at the next age (quantifying indirect genetic effects from the phenotype). The genotype is assumed to be developmentally independent (that is, controls **y** are open-loop), which means that there is no arrow towards the genotype. This diagram is a simplification of that considered by the evo-devo dynamics framework[59], so the brain model can be extended and the framework can still be used to analyse it.

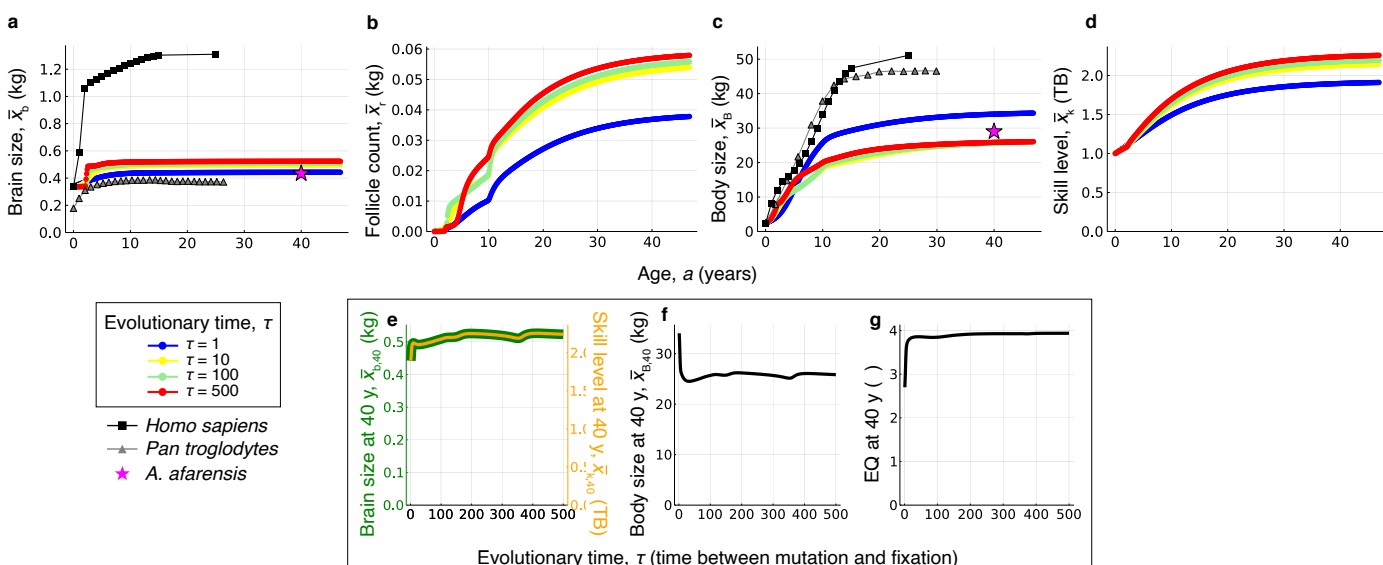

**Extended Data Fig. 2 | Evo-devo dynamics of brain size under *afarensis* scenario. a-d**, Developmental dynamics over age (horizontal axis) and evolutionary dynamics over evolutionary time (differently coloured dots; bottom left label): Evo-devo dynamics of **a**, brain size; **b**, follicle count; **c**, body size; and **d**, skill level. Evolutionary dynamics of (**e**, green) brain size, (**e**, orange) skill level, (**f**) body size, and (**g**) encephalisation quotient (EQ) at 40 years of age. **a,c**, The mean observed values in a modern human female sample are shown in black squares (data from ref. 39 who fitted data from ref. 69). One evolutionary time unit is the time from mutation to fixation.

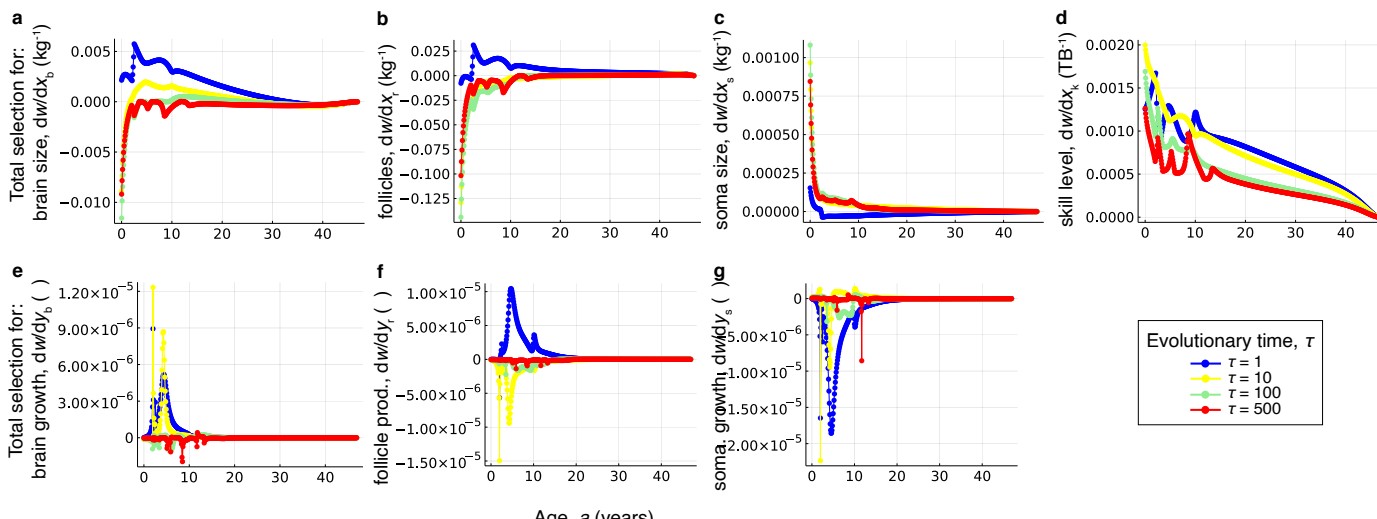

**Extended Data Fig. 3 | The action of total selection. a-d**, Total selection on brain size, follicle count, somatic tissue size, and skill level at each age over evolutionary time. Total selection for skill level over life persists at evolutionary equilibrium (red dots in **d**). **e-g**, Total selection on effort for brain growth, follicle production, and somatic growth at each age over evolutionary time. Total selection for genotypic traits nearly vanishes at evolutionary equilibrium (red dots in **e-g**), indicating that a path peak on the fitness landscape is reached. All plots are for the *sapiens* trajectory of Fig. 2b.

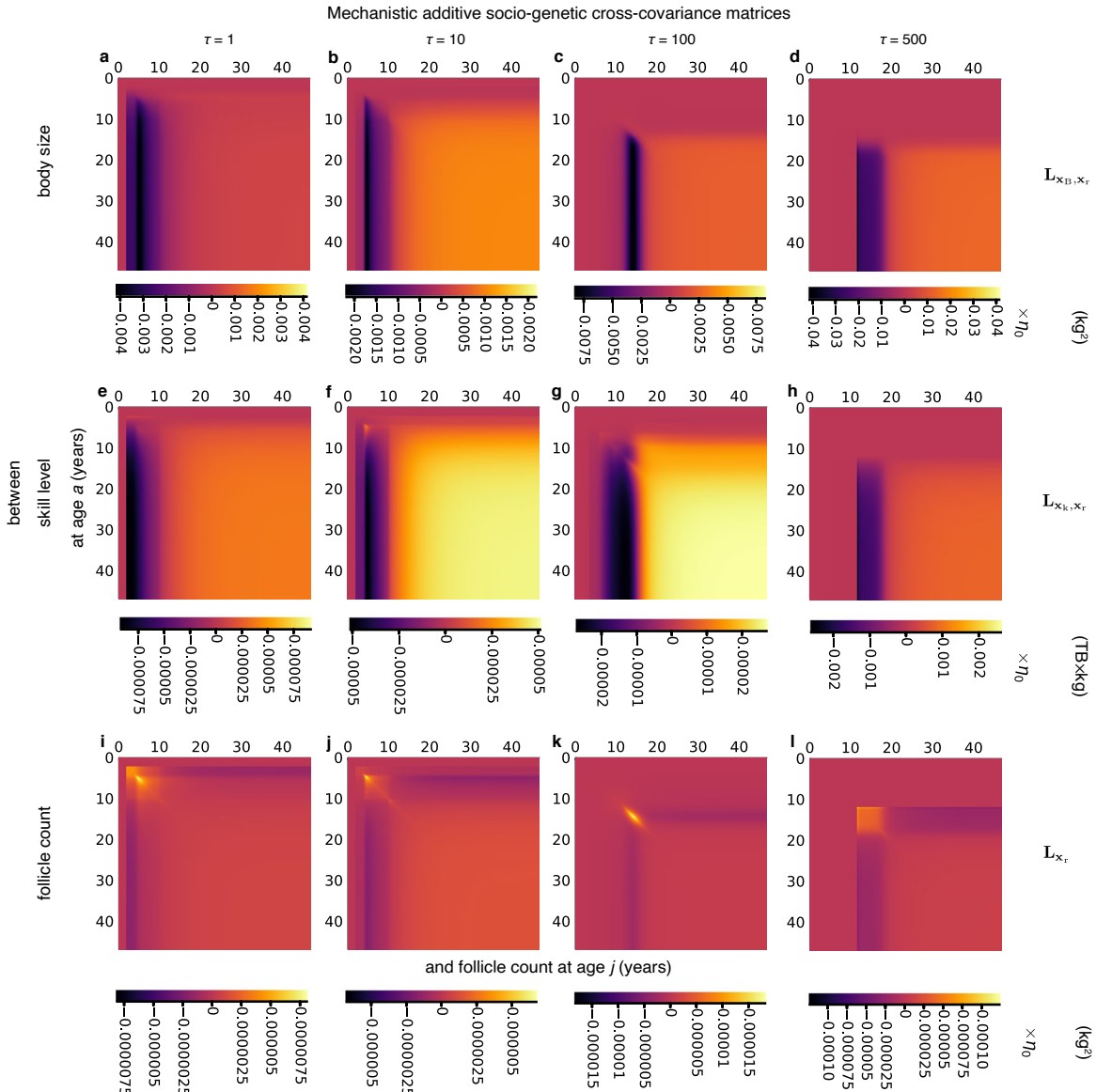

**Extended Data Fig. 4 | The action of constraint on body, skill, and follicle count expansion.** Mechanistic socio-genetic cross-covariance matrix between: **a-d**, body size and follicle count, **e-h**, skill level and follicle count, and **i-l**, follicle count and itself. All plots are for the sapiens trajectory of Fig. 2b.

# Reporting Summary

## Statistics

For all statistical analyses, confirm that the following items are present in the figure legend, table legend, main text, or Methods section.

| n/a | Confirmed | |
|---|---|---|
| ☐ | ☒ | The exact sample size (*n*) for each experimental group/condition, given as a discrete number and unit of measurement |
| ☒ | ☐ | A statement on whether measurements were taken from distinct samples or whether the same sample was measured repeatedly |
| ☒ | ☐ | The statistical test(s) used AND whether they are one- or two-sided *Only common tests should be described solely by name; describe more complex techniques in the Methods section.* |
| ☒ | ☐ | A description of all covariates tested |
| ☒ | ☐ | A description of any assumptions or corrections, such as tests of normality and adjustment for multiple comparisons |
| ☐ | ☒ | A full description of the statistical parameters including central tendency (e.g. means) or other basic estimates (e.g. regression coefficient) AND variation (e.g. standard deviation) or associated estimates of uncertainty (e.g. confidence intervals) |
| ☒ | ☐ | For null hypothesis testing, the test statistic (e.g. *F*, *t*, *r*) with confidence intervals, effect sizes, degrees of freedom and *P* value noted *Give P values as exact values whenever suitable.* |
| ☒ | ☐ | For Bayesian analysis, information on the choice of priors and Markov chain Monte Carlo settings |
| ☒ | ☐ | For hierarchical and complex designs, identification of the appropriate level for tests and full reporting of outcomes |
| ☒ | ☐ | Estimates of effect sizes (e.g. Cohen's *d*, Pearson's *r*), indicating how they were calculated |

*Our web collection on statistics for biologists contains articles on many of the points above.*

## Software and code

Policy information about availability of computer code

| Data collection | No data was collected in this study and so no software was used for data collection. |
|---|---|
| Data analysis | Custom computer code to run the model and generate the figures was prepared to run in Julia 1.7.2. All code is available in the Supplementary Information and in https://doi.org/10.5281/zenodo.10887414. |

For manuscripts utilizing custom algorithms or software that are central to the research but not yet described in published literature, software must be made available to editors and reviewers. We strongly encourage code deposition in a community repository (e.g. GitHub). See the Nature Portfolio guidelines for submitting code & software for further information.

## Data

Policy information about availability of data

All manuscripts must include a data availability statement. This statement should provide the following information, where applicable:

- Accession codes, unique identifiers, or web links for publicly available datasets
- A description of any restrictions on data availability
- For clinical datasets or third party data, please ensure that the statement adheres to our policy

No data was collected in this study. All data used were previously published in references provided in the main text or supplementary information.

## Human research participants

Policy information about <u>studies involving human research participants and Sex and Gender in Research.</u>

| | |
|---|---|
| Reporting on sex and gender | This study did not involve real human participants. The model considers only (simulated) females. |
| Population characteristics | This study did not involve real human participants. |
| Recruitment | This study did not involve real human participants. |
| Ethics oversight | This study did not involve real human participants. |

Note that full information on the approval of the study protocol must also be provided in the manuscript.

# Field-specific reporting

Please select the one below that is the best fit for your research. If you are not sure, read the appropriate sections before making your selection.

☐ Life sciences  ☐ Behavioural & social sciences  ☒ Ecological, evolutionary & environmental sciences

For a reference copy of the document with all sections, see <u>nature.com/documents/nr-reporting-summary-flat.pdf</u>

# Ecological, evolutionary & environmental sciences study design

All studies must disclose on these points even when the disclosure is negative.

| | |
|---|---|
| Study description | This study mathematically models the evolutionary and developmental dynamics of hominin brain and body size. |
| Research sample | This study did not collect data. It used previously published data of various kinds. First, brain and body sizes for human females over ontogeny were obtained from the spline fit of Kusawa et al (2014) to the cross-sectional data of Dekaban and Sadowsky (1978) for 1963 females from several hospitals in Washington, DC and Bethesda, MD. Second, fossil and extant data for adult brain and body sizes for 11 hominin species were taken from Smaers et al (2021) who took them from Grabowski et al (2015) and Kappelman (1996) among others. Fossil data for adult brain and body size for H. floresiensis were taken from Brown et al (2004) and for H. naledi from Garvin et al (2017). Fossil and extant data for adult brain and body sizes for great apes, lesser apes, cercopithecines, lemurs, and lorises were taken from Smaers et al (2021) who took them from multiple sources. Third, the mathematical model depends on various parameters that were estimated by González-Forero et al (2017) and González-Forero and Gardner (2018) from previously published metabolic, demographic, and neuroscience data with references in such publications. |
| Sampling strategy | This study did not collect data. |
| Data collection | This study did not collect data. |
| Timing and spatial scale | This study did not collect data. |
| Data exclusions | Failed organisms in Fig. 2a (gray dots) were not included in the calculation of the regression coefficient reported in that figure. The reason is that failed organisms here have a body composed entirely of brain and would thus be inviable. |
| Reproducibility | All attempts to repeat the model solutions yielded the results reported. |
| Randomization | This study did not collect data. |
| Blinding | The data used were blinded by the original publications where the data were taken from. |

Did the study involve field work?  ☐ Yes  ☒ No

# Reporting for specific materials, systems and methods

We require information from authors about some types of materials, experimental systems and methods used in many studies. Here, indicate whether each material, system or method listed is relevant to your study. If you are not sure if a list item applies to your research, read the appropriate section before selecting a response.

## Materials & experimental systems

| n/a | Involved in the study |
|-----|-----------------------|
| ☒ | Antibodies |
| ☒ | Eukaryotic cell lines |
| ☒ | Palaeontology and archaeology |
| ☒ | Animals and other organisms |
| ☒ | Clinical data |
| ☒ | Dual use research of concern |

## Methods

| n/a | Involved in the study |
|-----|-----------------------|
| ☒ | ChIP-seq |
| ☒ | Flow cytometry |
| ☒ | MRI-based neuroimaging |

