## [Peer Review File · Nature Human Behaviour]

Peer Review Information

Journal: Nature Human Behaviour

Manuscript Title: Evo-devo dynamics of hominin brain size

Corresponding author name(s): Mauricio González-Forero

Reviewer Comments & Decisions:

Decision Letter, initial version:

5th June 2023

Dear Dr González-Forero,

Thank you once again for your manuscript, entitled "Evo-devo dynamics of human brain size", and for your patience during the peer review process.

Your Article has now been evaluated by 4 referees. You will see from their comments copied below that, although they find your work of potential interest, they have raised quite substantial concerns. In light of these comments, we cannot accept the manuscript for publication, but would be interested in considering a revised version if you are willing and able to fully address reviewer and editorial concerns.

We hope you will find the referees' comments useful as you decide how to proceed. If you wish to submit a substantially revised manuscript, please bear in mind that we will be reluctant to approach the referees again in the absence of major revisions. If we send your revised manuscript out for re-review, we will also ask our peer reviewers to run and verify your code (please see our editorial on our code review process: <https://www.nature.com/articles/s41562-021-01190-w>). We are committed to providing a fair and constructive peer-review process. Do not hesitate to contact us if there are specific requests from the reviewers that you believe are technically impossible or unlikely to yield a meaningful outcome.

To guide the scope of the revisions, the editors discuss the referee reports in detail within the team, including with the chief editor, with a view to (1) identifying key priorities that should be addressed in revision and (2) overruling referee requests that are deemed beyond the scope of the current study. We hope that you will find the prioritised set of referee points to be useful when revising your study. Please do not hesitate to get in touch if you would like to discuss these issues further.

1) Our reviewers all indicate significant concerns about the validity (and clarity) of the assumptions and data that inform your model. Please address these in full, revising your approach as necessary.

2) Reviewer 1 points out that description is not explanation. We agree with this important point. Relatedly the word 'driver' (and related language) suggests mechanisms that are not clear nor supported by the evidence presented here, which Reviewer 4 also alludes to. Please ensure that the framing of (and discussion in) your manuscript is accurate with regard to what you can and cannot speak to with your current approach.

3) Reviewer 4 notes that there are differences in the observed data and model predictions in Figure 1h and 1o; please address these inconsistencies in your revised manuscript.

4) Please run and present sensitivity analyses as requested by Reviewer 1.

If you wish to submit a suitably revised manuscript, we would hope to receive it within 4 months. I would be grateful if you could contact us as soon as possible if you foresee difficulties with meeting this target resubmission date.

- Include a "Response to the editors and reviewers" document detailing, point-by-point, how you addressed each editor and referee comment. If no action was taken to address a point, you must provide a compelling argument. When formatting this document, please respond to each reviewer comment individually, including the full text of the reviewer comment verbatim followed by your response to the individual point. This response will be used by the editors to evaluate your revision and sent back to the reviewers along with the revised manuscript.
- Highlight all changes made to your manuscript or provide us with a version that tracks changes.

[REDACTED]

Thank you for the opportunity to review your work. Please do not hesitate to contact me if you have any questions or would like to discuss the required revisions further.

Sincerely,

[REDACTED]

Reviewer expertise:

Reviewer #1: evolution of hominin brain size

Reviewer #2: brain size evolution and development

Reviewer #3: bio anth, primate brain evolution

Reviewer #4: computational/mathematical modelling; human evolution

REVIEWER COMMENTS:

Reviewer #1:

Remarks to the Author:

This paper is a follow-up on the author's previous 2018 paper. It seeks to develop and extend an 'evo-devo' perspective on evolutionary processes, with a particular emphasis, as in the earlier paper, on brain evolution in the hominin lineage. It seeks to develop a model that combines both developmental processes and selection processes. While I have a lot of sympathy with the general approach, and especially the importance of incorporating developmental constraints into evolutionary arguments, this version suffers from several of the same conceptual flaws as the 2018 paper. In part, these seem to arise

from a deep philosophical confusion over the nature of biological explanation, and in part from a naïve understanding of primate (and hence hominin) sociality and the role this plays in their evolution. These sources of confusion are not helped by places where the writing is so opaque that it is simply not possible to figure out what is meant.

Major issues:

(1) When exploring evolutionary explanations, it is necessary to keep why? and how? explanations very carefully separated. Why questions, by definition, always have pre-eminence in biology, but we do want to know what is possible and what not – and that is why developmental/ mechanisms questions become important. Yes, they are both needed, but they play very different roles in the explanatory process. Confusing them is to commit what philosophers call a 'category mistake'.

(2) I would question the claim, repeated endlessly through the text, that the model tells us something new that contradicts what we previously thought was the case. The model actually tells us what we always knew (that there are constraints as well as selection factors at work, and they push and pull the organism in different ways). The problem has always been how to integrate the two components. This model doesn't solve that problem because, while it allows us to evaluate the developmental component, it unfortunately loses contact with the selection component at the same time. Identifying the rate of selection does not tell us why the selection pressure occurred -- although it might draw our attention to where to look.

(3) One of the risks we run in developing very complex models is that it is very easy to lose track of what is actually happening within the model, and hence to draw conclusions that are unwarranted – or, worse, simply wrong because we have not understood what has actually happened inside the model. This is commonly known as the GIGO (garbage in, garbage out) mode of modelling. The rule of thumb for good modelling is that if you cannot explain the model in a few simple words you probably haven't understood it yourself. The level of explanation given in the MS is such as to make me worry that the author actually doesn't really know what the model does or why, or what it actually means.

(4) This last issue is compounded by a common error, namely thinking that if you can formulate a problem as a model then you have explained the phenomenon. This is not so: a (mathematical) description is not an explanation. Sadly, too many people in too many disciplines seem to think it is, but that isn't an excuse. It risks misleading readers by reinforcing ideological biases under the guise of "being scientific".

(5) I am very sympathetic to modelling problems of evolution in this way, but I am concerned that the approach adopted here lacks the power that we really need. It is not enough just to explain how an effect was achieved developmentally. What we really want to know is what drove the evolutionary pathway in the direction it eventually took. And that needs a reverse engineering approach that integrates these processes with the environmental factors actually at work. And that in turn requires sensitivity analysis to determine the boundary conditions under which the predictions hold. A model that produces the same outcome no matter how it is tweaked explains nothing.

(6) There is a major confusion at the centre of the model, namely that of assuming that the principal options as selection factors explaining brain evolution are either ecological or social. This is a problem for two reasons. First, the original 2018 model confused the social explanations with Machiavellian Intelligence, and the current paper seems not to have learned the lesson of why this was wrong. It was/is nice to know that the MIT did not drive brain evolution (in primates or anyone else), but we could have told you that without having to go to all this effort. No one has believed the MIT since the 1990s: it is an emergent property of having a large brain and living in large groups, but it did not, and cannot, select for either. Indeed, at its heart it is destructive. The problem is that the model does not address the real social issue because that gets wrapped up with other ecological drivers. This is, admittedly, a common mistake, but that doesn't make it right to perpetuate the error. The selection factors are always ecological. The issue is how animals solve these ecological challenges (individually vs socially). Doing it socially requires additional cognitive skills. You are welcome to ignore these factors, but don't claim you have done a comprehensive model when you haven't.

(7) The assumption that cognitive skills can be measured independently in some way is not defensible. First, cognition and the brain are identical, so cognition cannot be meaningfully separated from the brain. Showing that brain and cognition are correlated in some way is simply circular reasoning: of course they are. Second, the assumption that all primate/hominin cognition involves is learning and memory flies in the face of everything we know about cognition and neurobiology: such an assumption might be OK for

voles, but it is certainly not OK for primates [or hominins]. Primates evolved a completely different form of cognition (see Passingham & Wise 2011): it involves manipulating information, not just remembering stuff. The evidence is quite clear and quite uncontroversial (as the MacLean et al. mega-study very clearly shows).

(8) There is a lot of definitional slippage that isn't helpful, and might be directly misleading. Using the term "driver" in respect of developmental processes in the evolutionary context may be common in the evo-devo context, but it risks seeming to put developmental processes on the same logical footing as selection processes. There is a sense in which that is true, but that is always internal to the organism. Development does influence evolution, it merely limits the range of possible pathways (Nick Davies's machine-gun-toting butterflies issue) – and that is not what we conventionally mean by the drivers of evolution. Definitional slippage confuses the naïve (that's precisely why people find Intelligent Design appealing) and risks causing unnecessary (and unhelpful) disputes.

Minor issues:

Throughout: the constant use of 'we' when it's a single author (and the reader isn't included) reads a bit oddly. It will be better as indirect speech: e.g. l. 107: "An overview...is given..." Otherwise, just use "I".

p. 1, Abstract, l.6: there is no evidence that 'culture' has selected for anything in brain evolution. I realise some folk are desperate to claim that it does, but at best it is a special case explanation since hardly any species have serious culture (or cultural transmission) in any meaningful sense. An N=1 does not make for a meaningful evolutionary explanation.

p. 1, abstract, l.7-8: the meaning of this claim is opaque, and in one sense meaningless. Selection can never be unconstrained, except in theory (in which case, the claim is necessarily wrong because this is in fact an assumption of the model).

l. 7: this is a very odd, and largely irrelevant, list of references: I am not sure any of them discuss the evolution of human brains, and many don't actually provide any meaningful evidence to support their claims (e.g. Humphreys, Henrich, Laland, Rosatti).

l. 8: This is a very odd set of citations for the claim that anyone is testing anything about brain evolution. [12] does not test brain evolution in humans, and anyway tests what constrains brain sizes, not what selects for brain evolution (a common error, and a sad reflection on referees from journals that ought to have known better); [13] and [14] do not test brain evolution in anything.

l. 13-14: how could you possibly test anything about the evolution of brains by manipulating something in any one species? These always have to be "postdictions" (as they are known in the philosophy of science), which is what your model rightly does.

l. 21: why are any of these papers actually relevant to testing hypotheses about brain evolution in humans? One is a model, and a model cannot test anything – it can only provide evidence of possibility; another claims to show that the Machiavellian Intelligence Hypothesis is true, but MIT is at best a consequence of having a large brain, not its cause. And culture doesn't explain anything – for the same reason (never mind the fact that the best it can do is plead a special case for one species – and spare me the absurd claims that chimpanzees have anything remotely close to meaningful culture).

Box 1: this is just a form of Waddington epigenetics. Credit should be given where credit is due.

Fig. 1 legend, l. 6 and l.283-4: EQ is, frankly, meaningless. It doesn't correlate with anything, and its widespread contemporary use completely misunderstands why Jerison introduced it. The real problem with it, as has been pointed out repeatedly, is that when EQ changes we don't actually know which component is responsible for the change. The ONLY thing that correlates with behaviour and cognition is absolute brain (or brain region) volume. That's because much of what is important (and allows animals to do clever stuff) is a massive (mainly neocortical) neural network that connects processing units in different parts of the cortex.

I. 228: how do we know the ancestral state lacked an adolescent growth spurt? Or, rather, to put it another way, when did it appear and why, and does the model predict that and tell us why?

I. 239: why does this model work better than others? We need an explanation, not simply a statement of fact. And I think we'd like to see some kind of sensitivity analysis so that we can be persuaded that it is only at certain parameter value sets that we get the 'right' prediction.

I. 293: really? Doesn't the model simply show what we already know, namely that selection is not free to run where it will?

I. 250-57: (a) It is not enough that the model happens to predict what we see. We want to know why it does so. (b) A bang-bang strategy is a description not an explanation.

I. 266-270: why? this is worrying, and doubly so when no attempt is made to deal with the problem.

I. 375-6: why would you need a big brain for cultural transmission? It's a cheap way of solving a 'how to' challenge. It doesn't involve anything especially complex. And no one has offered any concrete evidence that it is a driver [rather than a beneficial consequence] of evolving large brains. Hand-waving and enthusiasm are not evidence.

Fig. 4: In a sense, this figure encapsulates the central problem: the model seems to predict what is going on [the general trajectory], but we (a) don't really know why (or at least we are offered no explanation) and (b) we have no idea what particular environmental challenges were responsible (other than some vague arm-waving about environmental challenges).

(a) It is interesting to see that the two dominant 'forces' here are cooperation and ecological. That is exactly what the social brain hypothesis entails. Cooperation, by the way, is not about hunting (with the possible exception of Neanderthals); H. sapiens is a stealth hunter (using bows/arrows rather than spears) and that is best done alone or in very, very small groups – too many people disturb the prey. Nor is it about gathering. Doing anything in large groups is exceedingly challenging. Gathering is certainly done in bigger groups, but that is entirely driven by concerns over safety. In any case, the issue is not the evolution of cooperation (which comes as a free by-product of living in groups) but the much more taxing problem of the evolution of coordination (how to prevent groups dispersing).

(b) It is notable that the species where ecology dominates everything are precisely the three species that lived at high latitudes (erectus, Neanderthal and sapiens). High latitudes were extremely challenging both because they were colder [brain size decreases in these species as they move further north] and because animals become massively time-stressed [due to short day lengths in winter]. To cope with those challenges, these later species had to come up with some novel solutions.

Fig. 5 is simply incomprehensible, and the legend doesn't tell us anything meaningful.

I. 639-644: (a) the use of the term 'creative' here is misleading. (b) Development is not 'driving' anything, it is constraining in exactly the sense identified by Waddington. Unless, of course, you want to defend a Haeckelian view of evolution (evolution is the inevitable unfolding of a developmental pathway independently of the environment) – a view with a suspiciously strong undercurrent of Great-Chain-of-Being Lamarckism.

I. 691-6: this sounds suspiciously like an 'apples and oranges' problem: these aren't apples because I'm going to define them as oranges.

I. 706-10: rather grandiose? This is hardly a Higgs boson issue by any stretch of the imagination – not least because nothing in organismic biology allows us to make the kinds of 'strong predictions' that physics does.

Robin Dunbar

Reviewer #2:

Remarks to the Author:

I read and reviewed this paper as an evolutionary biologist interested in human brain evolution and not as a mathematician checking the equations. I can therefore contribute some general considerations that may be helpful when revising the manuscript. Overall, I find the research topic to be interesting and the findings intriguing. However, I believe that the issues mentioned below should be addressed in order to improve the clarity and accuracy of the paper.

For readers with a similar background to myself, it would be helpful if the model assumptions were presented more clearly in the opening section of the paper, rather than solely in the Methods section. Additionally, it would be beneficial to specify the data input into the model more explicitly.

I have reservations about the use of point estimates for body mass and brain mass to represent entire species while ignoring within-species variance. Furthermore, it is worth noting that some data represent species averages while others represent single individuals. Even when this issue about within-group variance is disregarded, it is important to acknowledge that for most of the fossil species, estimates of body size are imprecise approximations.

I would like to stress that the model's assumption (only casually mentioned in passing) that "brain size of a newborn is fixed and cannot evolve" is unrealistic, and at odds with the well documented prenatal growth differences between humans and apes, and the (admittedly sparse) fossil evidence from hominin endocasts. I appreciate that models have to make simplifying assumptions, however, this one seems problematic.

The term "human brain evolution" is used throughout the manuscript, but as the study includes several species that do not belong to the human genus, I would suggest using the more appropriate term "hominin brain evolution" instead. This would better reflect the scope of the study and avoid any potential confusion.

Regarding Box 1, I found the content to be confusing and suggest that it be removed from the manuscript entirely. While I appreciate the author's intention to provide additional context, the information presented does not seem to add clarity or depth to the paper's main focus.

In Figure 2 and the accompanying text, I recommend changing the phrase "Brain ... body size at 40 years of age" to "Adult brain and body size." It is highly unlikely that many (if any) of the hominin individuals mentioned in this figure lived to the age of 40. Additionally, it is unclear why the authors chose to model brain size until the age of 40 specifically, so this could be explained further in the text.

In the main text you mention that "a protracted human childhood arises from the trade-off of energy allocation between brain and somatic growth, so it is a consequence of brain expansion rather than being selected for." It would be helpful to discuss this with regards to suggestions that evidence for protracted brain growth can be observed in some Australopithecus species, which are generally considered to be rather ape-like in both endocranial volume and body mass.

I was also wondering how the small endocranial volumes of Homo naledi and especially Homo floresiensis (which were excluded from the regression model, if I understood correctly) can be explained in light of the author's model.

Please explain what you are trying to accomplish by modelling "failed organisms" with random growth efforts, and which aspects of your model/assumptions you are trying to probe or test.

Reviewer #3:

Remarks to the Author:

NATHUMBEHAV-23030879

Evo-devo dynamics of human brain size

This is an interesting manuscript that addresses a longstanding debate in evolutionary biology – whether social or ecological factors are primarily responsible for the evolution of the large human brain. The authors take a computational modelling approach that accounts for both evolutionary and developmental processes. I recommend major revisions to the manuscript, since clearer definitions would expand readership and the evolutionary mechanisms underlying their interpretations of the model are not clear. My comments are listed below:

Definitions:

What is “socio-genetic” covariation?

The author should provide clearer definitions throughout. A figure depicting the different variables and their interactions/effects would also be helpful.

Selection versus constraint:

This model will only be considered by evolutionary biologists if it reflects actual evolutionary mechanisms. As written, it is not clear that the scenarios presented in the text align with these mechanisms. Evolution is defined as a change in allele frequencies over generations – how exactly would the mechanisms/relationships described enact this?

For instance, the author suggests that constraint can drive evolutionary change and uses Box 1 as an example. However, it appears to be that while the path represents the constraint, the volcanic eruption could represent selection.

Also, the author suggests that ecology and culture drive human brain expansion in the model by affecting developmental/socio-genetic constraints rather than unconstrained selection. But how would ecological and cultural factors alter genetic covariation? Through what mechanism? There appears to be some fundamental relationship/interaction missing from this model. For instance, one major assumption upon which this model is built is that “direct selection on developmentally late reproductive tissue provides a force for reproductive tissue expansion, and socio-genetic covariation diverts this force to cause brain expansion.” How exactly would ecology and culture interact with reproductive tissue? Similarly, we know that fertility is not actually proportional to the size of reproductive tissue. What could this relationship represent in true evolutionary terms?

Minor points:

“We” versus “I” – perhaps use the latter for analyses conducted within this solo authored paper and the former for previous work by the author and their colleagues?

Lines 206, 214 – “consistent” instead of “consistently”

I hope the author finds my comments helpful in revising their manuscript.

Reviewer #4:

Remarks to the Author:

The specific reasons behind the evolution of the human brain to its current size remain uncertain. Despite various hypotheses and comparative studies, which have provided insightful findings, they are primarily qualitative and fail to fully explain the size of the human brain. In this paper, the authors take a mathematical approach to integrate the developmental and evolutionary dynamics of human brain size. Their research reveals that the expansion of the brain is a result of the interplay between ecological factors and cultural influences, leading to covariation between brain size and the size of late-developing

reproductive tissues.

This paper presents an intriguing perspective that sheds light on the understanding of the current size of the human brain. Besides, they hold great potential for elucidating the evolution of other organs, such as lungs and livers. However, I have a few suggestions to improve this manuscript.

1. In my view, constraints guide rather than drive evolution, with selection being the driving force. Using the example in Box 1, gravity drives the flow of mud downhill, while topographic constraints simply guide its movement.

2. The authors should refine the presentation of their results and consider omitting less important details. For instance, lines 126-129 mention that blue dots represent initial growth efforts, which are likely illustrated in Figure 1. Listing all 16 panels and explaining each one in the main text is unnecessary. Focusing on the main points and disregarding less relevant details or relocating them to the supplementary information (SI) section would help readers better grasp the intended message. Additionally, the use of four evolutionary time, τ , could be clarified in the caption of Figure 1 by explaining the meaning of the "horizontal axis in A."

3. Similarly, regarding the statement in lines 152-155, it is challenging to draw such a conclusion from Figure 1, which shows that "allocation to reproductive growth developmentally increases from zero after 3 years of age and slowly achieves a small maximum value at around 20 years of age." While there may be several conclusions, the data are not particularly remarkable. The authors should avoid making such unsupported statements.

4. Regarding Figure 1h, in addition to the developmental delay observed in the model compared to the actual data (red dots relative to black squares), there is also a significant difference in the developmental trend. The observed data depict rapid growth in the early stages followed by slower growth after approximately 3 years, whereas the model predicts a step-like change that lacks coherence. Similar inconsistencies can be observed in Figure 1o.

Author Rebuttal to Initial comments

Reviewer 1

Reviewer # 1:

Remarks to the Author:

This paper is a follow-up on the author's previous 2018 paper. It seeks to develop and extend an 'evo-devo' perspective on evolutionary processes, with a particular emphasis, as in the earlier paper, on brain evolution in the hominin lineage. It seeks to develop a model that combines both developmental processes and selection processes.

I thank the reviewer for their detailed comments. I have sought to address them thoroughly as described below.

While I have a lot of sympathy with the general approach, and especially the importance of incorporating developmental constraints into evolutionary arguments, this version suffers from several of the same conceptual flaws as the 2018 paper. In part, these seem to arise from a deep philosophical confusion over the nature of biological explanation, and in part from a naïve understanding of primate (and hence hominin) sociality and the role this plays in their evolution. These sources of confusion are not helped by places where the writing is so opaque that it is simply not possible to figure out what is meant.

I have reformulated the presentation to reduce the scope for philosophical disagreement. Major issues:

(1) When exploring evolutionary explanations, it is necessary to keep why? and how? explanations

very carefully separated. Why questions, by definition, always have pre-eminence in biology, but we do want to know what is possible and what not – and that is why developmental/mechanisms questions become important. Yes, they are both needed, but they play very different roles in the explanatory process. Confusing them is to commit what philosophers call a ‘category mistake’.

I have reformulated the presentation to reduce the scope for philosophical disagreement.

(2) I would question the claim, repeated endlessly through the text, that the model tells us something new that contradicts what we previously thought was the case.

I have removed the novelty claims as indicated by the additional instructions to authors for *Nature Human Behaviour* manuscripts that were attached in the decision email.

The model actually tells us what we always knew (that there are constraints as well as selection factors at work, and they push and pull the organism in different ways).

To clarify the main point of the manuscript, I have rewritten the sentences in the abstract describing the main result: ‘*Analysis shows that in this model the brain expands because it is “socio-genetically” correlated with developmentally late preovulatory ovarian follicles, not because brain size is directly selected for. The socio-genetic correlation causing the recovered hominin brain expansion is generated over development by ecology and possibly culture.*’ None of the main points of the manuscript is that “there are constraints as well as selection factors at work, and they push and pull the organism in different ways”.

The problem has always been how to integrate the two components. This model doesn’t solve that problem because, while it allows us to evaluate the developmental component, it unfortunately loses contact with the selection component at the same time. Identifying the rate of selection does not tell us why the selection pressure occurred – although it might draw our attention to where to look.

It is unclear what the reviewer means by “los[ing] contact with the selection component”. None of the main points of the manuscript is to identify “the rate of selection”. Perhaps one could say that the model “loses contact with the selection component” in that the model assumes a simple fitness function that does not directly depend on many of the things that have been hypothesised, such as by survival depending on skills and ecological and social challenges. Yet, the model is as simple as possible while still being able to obtain quantitative predictions and recover at least human-sized brains and bodies. With those objectives it was not necessary to make “the selection component” any more complex, although the model can be modified to do it. The last paragraph of the discussion now indicates how future work can compare this model with such possible extensions to identify models best explaining the data.

(3) One of the risks we run in developing very complex models is that it is very easy to lose track of what is actually happening within the model, and hence to draw conclusions that are unwarranted – or, worse, simply wrong because we have not understood what has actually happened inside the model.

Consider three possibilities: no model, a simple model that makes qualitative predictions, and a complex model that makes quantitative predictions. With no model, we have nature to understand, and for hominin brain expansion, it means we cannot do interventions and so causal understanding is severely limited in this option. With a simple model that makes qualitative predictions, we could only understand what causes the evolution of larger brains, but not what causes the evolution of human-sized brains. With a complex model that makes quantitative predictions, we could understand what causes the evolution of human sized brains, even if it takes more work than with a simple model. This is possible because a complex model still allows one to implement interventions and so the causes of the patterns it generates can still be understood, even if it takes more work.

If a simple model can make quantitative predictions mechanistically, then all the better, but that remains unavailable. In this situation, a complex model is better than no model. The final paragraph of the discussion now states that other models are welcome, more so if they can be simpler while also providing quantitative predictions mechanistically.

Now, if one has a complex model, tools that enable one to better understand the model would be helpful. The evo-devo dynamics framework allows for this, by enabling analytical treatment of a complex model such as the brain model. Analytical treatment is the reason that simple models are particularly understandable. Analogously, the application here of the evo-devo dynamics framework to the brain model gives precisely deeper understanding of such a complex model by using the new analytical machinery made available, an understanding that is not available with the dynamic optimization approach used before.

This is commonly known as the GIGO (garbage in, garbage out) mode of modelling.

GIGO means that garbage input produces garbage output. GIGO does not mean to draw conclusions that are unwarranted or wrong. A key aim of the present manuscript is to provide analytical machinery to understand a complex model such as the brain model. So this manuscript provides tools to reduce the risk of drawing unwarranted or wrong conclusions.

The rule of thumb for good modelling is that if you cannot explain the model in a few simple words you probably haven't understood it yourself. The level of explanation given in the MS is such as to make me worry that the author actually doesn't really know what the model does or why, or what it actually means.

The fourth paragraph of the introduction explains the model in a few sentences. By including development and evolution, the model yields many predictions and the original submission included excessively detailed descriptions of such predictions. I have now removed most of those descriptions to streamline the presentation. I have kept only the key descriptions and made them short.

- (4) This last issue is compounded by a common error, namely thinking that if you can formulate a problem as a model then you have explained the phenomenon. This is not so: a (mathematical) description is not an explanation. Sadly, too many people in too many disciplines seem to think it is, but that isn't an excuse. It risks misleading readers by reinforcing ideological biases under the guise of "being scientific".

Replicating a pattern is not explaining a pattern. But it is a key first step with the approach used here, and indeed across physics where this approach borrows inspiration from. One first has to replicate the pattern in silico to manipulate the pattern and identify its causes, given that one cannot do it in vivo. Accordingly, the approach here seeks to replicate patterns, and then manipulate them to identify what factors cause them.

- (5) I am very sympathetic to modelling problems of evolution in this way, but I am concerned that the approach adopted here lacks the power that we really need. It is not enough just to explain how an effect was achieved developmentally. What we really want to know is what drove the evolutionary pathway in the direction it eventually took. And that needs a reverse engineering approach that integrates these processes with the environmental factors actually at work. And that in turn requires sensitivity analysis to determine the boundary conditions under which the predictions hold. A model that produces the same outcome no matter how it is tweaked explains nothing.

As explained above, Fig. 2b contains a sensitivity analysis that shows "what drove the evolutionary pathway in the direction it eventually took". As further detailed above, a wide array of other sensitivity analyses are done elsewhere in the manuscript and in previous papers.

- (6) There is a major confusion at the centre of the model, namely that of assuming that the principal options as selection factors explaining brain evolution are either ecological or social.

The contrast between ecological and social hypotheses to explain brain size evolution is widespread, and endorsed by the reviewer as indicated by the following quote from the perhaps key paper where the reviewer's hypothesis is discussed: "This leaves us with just two classes of hypotheses, the ecological and the social" (Dunbar 1998 The Social Brain Hypothesis).

What the reviewer probably means is that in their hypothesis, the challenges proposed to drive brain

expansion are ecological but solved with social partners. So the contrast the reviewer seems to endorse is between ecological and cooperative challenges in the terminology of this manuscript. That is, ecological challenges are posed by the ecology and solved individually, whereas cooperative challenges are posed by the ecology and solved socially, but both are challenges posed by the ecology. This is explicitly considered in the model.

This is a problem for two reasons. First, the original 2018 model confused the social explanations with Machiavellian Intelligence, and the current paper seems not to have learned the lesson of why this was wrong.

The original 2018 model did not equate social explanations with Machiavellian intelligence. The 2018 model separately considered competition, which is a key aspect of the Machiavellian intelligence hypothesis, and cooperation, which is an aspect often emphasized by the social brain hypothesis. But there are overlapping elements between the two hypotheses as described below, since the social brain hypothesis, at least in one of its interpretations, has elements of the Machiavellian intelligence hypothesis.

It was/is nice to know that the MIT did not drive brain evolution (in primates or anyone else), but we could have told you that without having to go to all this effort. No one has believed the MIT since the 1990s: it is an emergent property of having a large brain and living in large groups, but it did not, and cannot, select for either. Indeed, at its heart it is destructive.

As the reasoning was verbal and the evidence correlational at the time, the statement that “we could have told you that without having to go to all this effort” would have reduced to an opinion, and others had other opinions. It is incorrect that “No one has believed the MIT since the 1990s”. The role of competition as driving cognitive arms races has remained prevalent after that, including in explanations of why humans evolved extraordinary intelligence (e.g., Flinn et al 2005, “Ecological dominance, social competition, and coalitionary arms races: Why humans evolved extraordinary intelligence” cited over 500 times now; <https://www.sciencedirect.com/science/article/pii/S1090513804000595>).

The problem is that the model does not address the real social issue because that gets wrapped up with other ecological drivers. This is, admittedly, a common mistake, but that doesn't make it right to perpetuate the error. The selection factors are always ecological. The issue is how animals solve these ecological challenges (individually vs socially). Doing it socially requires additional cognitive skills. You are welcome to ignore these factors, but don't claim you have done a comprehensive model when you haven't.

It is unclear what “the real social issue [...] gets wrapped up with other ecological drivers” means. The reviewer states “The issue is how animals solve these ecological challenges (individually vs socially)”. The 2018 model, as the one here, explicitly allows for individuals to overcome ecological challenges both individually and socially. The model calls these types of challenges ecological (or me vs nature) and cooperative (or us vs nature). The model additionally considers between-individual-competitive (or me vs you) and between-group-competitive (or us vs them). The cooperative challenges (us vs nature) correspond to the ones the reviewer seems to indicate are not present.

The reviewer's statement that “doing it socially requires additional skills” indicates that what the reviewer means is not merely that ecological challenges are solved socially, but *how* ecological challenges are solved socially. As stated in the reviewer's papers, the “additional skills” the reviewer has in mind as required by solving problems socially are, for instance, mind reading and self-control to keep social partners happy to cooperate. These “additional skills” are only needed if social partners have different interests that must be managed to keep partners happy. So, what the reviewer seems to mean is not that ecological problems are solved socially, but that they are solved socially and that there is a conflict of interests and that the conflict of interests is managed, which thus overlaps with the Machiavellian intelligence hypothesis. Thus, although the model already includes conflict, what the reviewer seems to have in mind is an interaction between cooperation and conflict, which is an additional layer of complexity that is not included in the model. Yet, this additional layer of complexity has not been necessary to obtain the evolution of human brain and body sizes in the model. This does not mean that this additional layer of complexity does not yield a better fitting model, but it was not necessary to evolve human-sized brains as proposed by the

hypothesis. This illustrates why it is key to have models making quantitative predictions. Without quantitative predictions, one cannot realise that the hypothesis despite its intuitive appeal is not necessary to evolve human-sized brains, at least given the data used for parameter values.

Now, as explained in the last paragraph of the discussion, the model shows how human brain size could have evolved but not how it actually evolved, and future elaborations and model selection techniques could make progress toward that goal. Such future elaborations may find a role for the reviewer's hypothesis for instance if the indicated additional layer of complexity is found to be part of models better explaining the data.

To clarify what the brain model includes, I expanded a description of this in the third paragraph of the introduction. I added a sentence at the end of this paragraph stating that the model incorporates basic aspects of leading hypotheses, but doesn't incorporate every aspect and can be modified to include them.

(7) The assumption that cognitive skills can be measured independently in some way is not defensible. First, cognition and the brain are identical, so cognition cannot be meaningfully separated from the brain. Showing that brain and cognition are correlated in some way is simply circular reasoning: of course they are.

It is unclear what the reviewer means by "measured independently". Tests of cognitive ability are routinely given to people of all ages, usually without measuring their brain size. Needless to say, the results of these tests are not identical to people's brain size, and the correlation between cognitive ability and brain size in humans is far from having $R^2 = 1$. The brain model does not show that brain and cognition are "correlated", but that in the model they have a mechanistic, quantitative relationship given by an equation derived from energy conservation. The distinction between correlation and mechanism is important because correlation lacks information about mechanism and causation.

Second, the assumption that all primate/hominin cognition involves is learning and memory flies in the face of everything we know about cognition and neurobiology: such an assumption might be OK for voles, but it is certainly not OK for primates [or hominins]. Primates evolved a completely different form of cognition (see Passingham & Wise 2011): it involves manipulating information, not just remembering stuff. The evidence is quite clear and quite uncontroversial (as the MacLean et al. mega-study very clearly shows).

The model has not explicitly modelled information manipulation for simplicity. The model is as complex as needed to make quantitative predictions, but it as simple as possible in this and other regards as a first approximation. As stated above, including this other additional layer of complexity was not necessary to evolve human-sized brains in the model. As described at the end of the discussion, the model can be expanded or other models can be built, and model selection methods can be used to identify models that best explain the data. This is left for future work.

(8) There is lot of definitional slippage that isn't helpful, and might be directly misleading.

I have now included definitions for each non-standard term, such as socio-genetic covariation and mechanistic additive genetic covariance.

Using the term "driver" in respect of developmental processes in the evolutionary context may be common in the evo-devo context, but it risks seeming to put developmental processes on the same logical footing as selection processes.

I have removed the term "driver" everywhere in the manuscript. The introduction now explains how the approach presented here can identify causes, by enabling manipulative experiments *in silico*. Accordingly, I now use the term "cause" consistently accompanied by the qualifier "in the model", to note that this approach identifies the causes of patterns *in* the model, not necessarily in reality.

The distinction between evolutionary and developmental causes is a philosophical matter that is based on the mathematical theory of the modern synthesis that ignored development. Specifically, that theory entails that evolutionary outcomes are defined only by selection. That is, adaptive evolution stops at peaks of the fitness landscape. All development does in that view is to modulate which outcome or peak is achieved, so the evolutionary role of development has come to be interpreted as described by the reviewer.

The evo-devo dynamics framework formulates mathematical theory that explicitly considers development and shows that evolutionary outcomes are not defined by selection alone, but by both development and selection. That is, adaptive evolution stops at path peaks on the fitness landscape, where the path is given by development. Such path peaks are generally not peaks on the landscape. Thus, the causes of adaptation, that is why the population is at a path peak, are both due to selection and development, even if there is a single peak on the landscape, or no peak as in the present manuscript (Fig. 5). Yet, this paper does not aim to dwell in philosophical issues and I have removed the volcano analogy and associated content to reduce the scope for philosophical disagreement.

There is a sense in which that is true, but that is always internal to the organism. Development does influence evolution, it merely limits the range of possible pathways (Nick Davies's machinegun-toting butterflies issue) – and that is not what we conventionally mean by the drivers of evolution.

As explained above, the statement that “Development [...] merely limits the range of possible pathways” is incorrect. Development defines jointly with the selection the evolutionary outcomes. This is a statement that is easily dismissed as being the same given that verbal descriptions are open for interpretation, but it is crucial to understand they are not the same. To see why consider the following example. In this manuscript, it is shown that changing development alone without changing (direct) selection, the brain and body sizes of seven different hominins evolve. In this example, development did not “merely limit the range of possible pathways”. It jointly defined the outcomes with selection, namely, the brain and body sizes of seven hominins.

Definitional slippage confuses the naïve (that's precisely why people find Intelligent Design appealing) and risks causing unnecessary (and unhelpful) disputes.

I have reformulated the presentation to reduce the scope for philosophical disagreement. Minor issues:

Throughout: the constant use of ‘we’ when it's a single author (and the reader isn't included) reads a bit oddly. It will be better as indirect speech: e.g. l. 107: “An overview...is given...” Otherwise, just use “I”.

I changed the presentation to first person singular or passive voice.

p. 1, Abstract, l.6: there is no evidence that ‘culture’ has selected for anything in brain evolution. I realise some folk are desperate to claim that it does, but at best it is a special case explanation since hardly any species have serious culture (or cultural transmission) in any meaningful sense. An N=1 does not make for a meaningful evolutionary explanation.

This manuscript does not show that “culture has selected for anything”. In this manuscript, culture is shown to affect development, without affecting (direct) selection. Explaining the evolution of a trait of a single species is the objective of this manuscript. The reviewer considers that as not being a meaningful evolutionary explanation as the reviewer thinks about evolution in comparative terms but that is because of the research tools they have had available. This manuscript presents tools that allow identifying evolutionary causes for the traits of a single species.

p. 1, abstract, l.7-8: the meaning of this claim is opaque, and in one sense meaningless. Selection can never be unconstrained, except in theory (in which case, the claim is necessarily wrong because this is in fact an assumption of the model).

I removed the term “unconstrained” throughout and now only use the term “direct” which is standard. The meaning of this claim is now made clearer by the new Fig. 5, which illustrates the fitness landscape of the model, showing that selection in the model only favours increased follicle count, but not larger brains. The meaning of “direct” selection is made clear by the new Extended Data Fig. 1, which shows that there is only one direct arrow to fitness, that from follicle count.

l. 7: this is a very odd, and largely irrelevant, list of references: I am not sure any of them discuss the evolution of human brains, and many don't actually provide any meaningful evidence to support their

claims (e.g. Humphreys, Henrich, Laland, Rosatti).

I have added 10 references, particularly focused on human brain evolution. I kept the references I had for the following reasons. Henrich and Laland are book-length treatments of cultural hypotheses intending to explain human brain evolution. Humphrey, Clutton-Brock & Harvey, Byrne & Whiten, Dunbar and Rosatti are key references introducing or reviewing ecological and social hypotheses. The sentence where these references are cited is not about evidence but about the hypotheses.

Yet, this list of references is not exhaustive as that is not the aim here.

l. 8: This is a very odd set of citations for the claim that anyone is testing anything about brain evolution. [12] does not test brain evolution in humans, and anyway tests what constrains brain sizes, not what selects for brain evolution (a common error, and a sad reflection on referees from journals that ought to have known better); [13] and [14] do not test brain evolution in anything.

This sentence is about tests of hypotheses of brain evolution and the citations are for reviews on tests including contrasting view points. Specifically, DeCasien et al is a recent review of correlative analyses of primate brain size evolution aiming to cover humans, but it does not test anything as it is a review. Hooper et al is a recent analysis questioning the ability of correlative analysis to yield insight into brain size evolution. Healy is a book-length review of tests of cognition evolution criticizing the interest on brain size, so it is cited to point to alternative view points.

l. 13-14: how could you possibly test anything about the evolution of brains by manipulating something in any one species? These always have to be “postdictions” (as they are known in the philosophy of science), which is what your model rightly does.

With experimental evolution one can manipulate the evolution of a single species. For instance, changing the selection pressure in the lab one can see the brain size that evolves, as Alexander Kotrschal and colleagues do in fishes. The approach here does similarly but in silico. It yields both postdictions and predictions (e.g., for the suite of challenge types faced by each species, or that *A. afarensis* had mild indeterminate body growth; Extended Data Fig. 2c).

l. 21: why are any of these papers actually relevant to testing hypotheses about brain evolution in humans? One is a model, and a model cannot test anything – it can only provide evidence of possibility; another claims to show that the Machiavellian Intelligence Hypothesis is true, but MIT is at best a consequence of having a large brain, not its cause. And culture doesn't explain anything – for the same reason (never mind the fact that the best it can do is plead a special case for one species – and spare me the absurd claims that chimpanzees have anything remotely close to meaningful culture).

The sentence with these citations is about models that yield qualitative predictions for brain evolution, not about empirical tests of the hypotheses. The cited papers are models that yield qualitative predictions for brain evolution.

Box 1: this is just a form of Waddington epigenetics. Credit should be given where credit is due.

Waddington epigenetics did not make the connection to evolution as given in this figure, specifically in that development imposes necessarily hard constraints to adaptation. In Waddington and until recently, the question of whether development imposes soft or hard constraints on adaptation was open.

To reduce the scope for philosophical disagreement and to improve the clarity of what the model shows, I removed this figure and included a new Fig. 5 that shows the fitness landscape of the brain model.

Fig. 1 legend, l. 6 and l.283-4: EQ is, frankly, meaningless. It doesn't correlate with anything, and its widespread contemporary use completely misunderstands why Jerison introduced it. The real problem with it, as has been pointed out repeatedly, is that when EQ changes we don't actually know which component is responsible for the change. The ONLY thing that correlates with behaviour and cognition is absolute brain (or brain region) volume. That's because much of what is important (and allows animals to do clever stuff) is a massive (mainly neocortical) neural network that connects processing units in different parts of the cortex.

This thinking is based on a regression based understanding of brain evolution. The meaning of EQ does not come from whether or not it correlates with other variables, but from the equation that defines it; that is, it is brain size relative to the expected brain size for a given body size. In a regression description, “we don’t actually know which component is responsible for the change”. But the model here is not correlational, and in it we do know what component is responsible for its change, as shown in the Fig. 3e-g. Change in EQ in the model is due to both change in brain and body size.

l. 228: how do we know the ancestral state lacked an adolescent growth spurt? Or, rather, to put it another way, when did it appear and why, and does the model predict that and tell us why?

I removed the discussion of growth spurts for lack of space.

l. 239: why does this model work better than others? We need an explanation, not simply a statement of fact. And I think we’d like to see some kind of sensitivity analysis so that we can be persuaded that it is only at certain parameter value sets that we get the ‘right’ prediction.

I removed this discussion, which was about other models of growth spurts, for lack of space. The sensitivity analyses have been done elsewhere and in this manuscript as explained above.

l. 293: really? Doesn’t the model simply show what we already know, namely that selection is not free to run where it will?

I removed the statement of novelty as indicated by the additional instructions to authors for *Nature Human Behaviour* manuscripts that were attached in the decision email. Yet, the model does not “simply show [...] that selection is not free to run where it will”. In particular, regarding the point made in the statement removed, the model shows that the expansion in EQ is caused by a change in development, not (direct) selection (see the new Fig. 2b).

l. 250-57: (a) It is not enough that the model happens to predict what we see. We want to know why it does so. (b) A bang-bang strategy is a description not an explanation.

I removed this discussion of growth spurts for lack of space. Yet, in other places of the manuscript, I have clarified why the model generates the patterns that it does.

l. 266-270: why? this is worrying, and doubly so when no attempt is made to deal with the problem.

This refers to the found sequence of menarche and then growth spurt, whereas girls typically (but not always) show the reverse and the correct girl sequence was found in the previous dynamic optimization approach (González-Forero- & Gardner 2018). In the original submission, I suggested that the reversed pattern was perhaps due to the ancestral genotypic traits used; the dynamic optimization approach did not have ancestral genotypic traits but “initial guesses” that play an analogous role and those were chosen by sequentially solving the model from an arbitrary starting point. In the original submission of the present paper, the ancestral genotypic traits were manually and so subjectively chosen. Since I find that genotypic traits may affect the evolved ontogenetic patterns, in this revision I identify the ancestral genotypic traits by running the model first under the parameter values that yield the evolution of brain and body sizes of *A. afarensis*. This approach is more realistic than that in the dynamic optimization approach and, relatively to the approach in the original submission, it removes subjectivity, slightly reduces the error in the predicted patterns, and gives new insights. Yet, the reverse girl sequence remains.

This result is now succinctly mentioned at the end of the fourth paragraph of the section Evo-devo dynamics of brain size. The six paragraph of the same section explains why the ontogenetic patterns in this evo-devo dynamics model may be delayed and different relative to those found in the previous optimisation approach, in particular due to age discretisation and choice of ancestral genotypes. Additionally, the final paragraph of the discussion acknowledges this fact by stating that the model does not perfectly recover the data, particularly for the ontogenetic patterns, and that means there is scope for future models or modifications of this model to improve predictions.

l. 375-6: why would you need a big brain for cultural transmission? It’s a cheap way of solving a ‘how to’ challenge. It doesn’t involve anything especially complex. And no one has offered any concrete ev-

idence that it is a driver [rather than a beneficial consequence] of evolving large brains. Hand-waving and enthusiasm are not evidence.

The introduction and start of the first result section now explain more clearly the role of culture. It is not that a big brain is needed for cultural transmission. Instead, it is that culture may in principle cause weakly rather than strongly diminishing returns of learning, by enabling individuals to continue to learn from accumulated knowledge in the population. Such weakly diminishing returns are found to be necessary to evolve human-sized brains in the model.

The traditional reasoning that large brains evolve because of cognitively challenging tasks, as illustrated by the reviewer's remark, is not found to be reliable to identify why human-sized brains evolve in the model. Indeed, culture is not needed because "it involves anything specially complex", nor management of social relationships is needed because it is more complex than foraging.

Fig. 4: In a sense, this figure encapsulates the central problem: the model seems to predict what is going on [the general trajectory], but we (a) don't really know why (or at least we are offered no explanation) and (b) we have no idea what particular environmental challenges were responsible (other than some vague arm-waving about environmental challenges).

To clarify why, I have rearranged the presentation and included a sensitivity analysis. Specifically, the figure mentioned by the reviewer is now Fig. 1 (it was Fig. 4 in the original submission), and now includes an additional result for the *afarensis* scenario; the first result section where this figure is presented describes the conditions that cause the evolution of brain and body sizes of seven hominins in the model. The following result section (Emergence of homin brain-body allometry) shows in Fig 2b that changing from one condition to the other moves from the evolution of australopithecine brain sizes to modern human brain sizes. The combination of Figs 1 and 2b shows the changes involved: in challenge proportions and the shape of energy extraction efficiency. This shows why the brain expansion happened in the model: because of an increase in ecological challenges and a shift to weakly diminishing returns of learning. Identifying the particular environments that could have been responsible for such changes is beyond the scope of this manuscript. One way to do this is to estimate the shape of energy extraction efficiency in hunter gatherers living in different environments. This is beyond the scope of this manuscript.

(a) It is interesting to see that the two dominant 'forces' here are cooperation and ecological. That is exactly what the social brain hypothesis entails.

Indeed, cooperation and ecology are the dominant challenge types in this reconstruction of hominin brain expansion. However, as shown in Fig. 3a,b of the 2018 paper, while the contribution of ecology is to increase brain size, the contribution of cooperation is to decrease brain size. This not what the social brain hypothesis proposes.

To remind the reader of this, this is now briefly recapped in the introduction (lines 56-58).

Cooperation, by the way, is not about hunting (with the possible exception of Neanderthals); *H. sapiens* is a stealth hunter (using bows/arrows rather than spears) and that is best done alone or in very, very small groups – too many people disturb the prey. Nor is it about gathering. Doing anything in large groups is exceedingly challenging. Gathering is certainly done in bigger groups, but that is entirely driven by concerns over safety. In any case, the issue is not the evolution of cooperation (which comes as a free by-product of living in groups) but the much more taxing problem of the evolution of coordination (how to prevent groups dispersing).

(b) It is notable that the species where ecology dominates everything are precisely the three species that lived at high latitudes (*erectus*, Neanderthal and *sapiens*). High latitudes were extremely challenging both because they were colder [brain size decreases in these species as they move further north] and because animals become massively time-stressed [due to short day lengths in winter]. To cope with those challenges, these later species had to come up with some novel solutions.

There is the issue that at least *sapiens* did not originate at high latitudes, and it is its origin that is the

concern here. The challenging ecology might come from different aspects of the ecology, and for *sapiens* in particular, it might have come from the harsh rift valley, which of course was faced by every other species living there, but the key requirement of weakly diminishing returns of learning (due to culture?) may not have been met in other species.

Fig. 5 is simply incomprehensible, and the legend doesn't tell us anything meaningful.

I moved 3/4 of the panels to what is now Extended Data Fig. 4 to simplify the figure. I also expanded the legend to explain what the figure shows.

l. 639-644: (a) the use of the term 'creative' here is misleading.

I replaced "creative" with "generative", which is hopefully less misleading while still conveying the intended meaning in a single word.

(b) Development is not 'driving' anything, it is constraining in exactly the sense identified by Waddington.

I removed the term "driving" everywhere. But development is not "constraining in exactly the sense identified by Waddington". Waddington did not show that development imposes necessarily hard constraints on evolution. It is precisely those hard constraints that matter here. Those hard constraints mean that changes in development cause different hominin brain and body sizes to evolve in the model. Again, it is only development, not direct selection, that is being changed in Fig. 1, where the brain and body sizes of seven different hominins evolve.

Unless, of course, you want to defend a Haeckelian view of evolution (evolution is the inevitable unfolding of a developmental pathway independently of the environment) – a view with a suspiciously strong undercurrent of Great-Chain-of-Being Lamarckism.

As explained above, I have removed 'drive' and 'creative' with more neutral, less teleological terms.

l. 691-6: this sounds suspiciously like an 'apples and oranges' problem: these aren't apples because I'm going to define them as oranges.

This text was removed for lack of space. The reviewer refers to a statement that said "We find that such costs are not fitness costs in the model, but instead affect mechanistic socio-genetic covariation". Although this was deleted, similar statements occur elsewhere. For instance, the second to last paragraph of the introduction ends by saying: "This mechanistic treatment shows that brain metabolic costs in the model are not direct fitness costs but affect mechanistic socio-genetic covariation". Yet, the remaining statements strictly use the adjective "direct". Direct fitness costs have a standard mathematical definition in quantitative genetics. I use such standard definition.

Now, common parlance of brain metabolic costs having fitness costs may be argued to refer to total fitness costs of brain metabolic costs, rather than direct fitness effects. To consider that possibility, I have included the found values for the total fitness effects of metabolic costs, which I find to be a often total fitness costs and but occasionally total fitness benefits (lines 452-454). I included a figure of this (Fig. S12). I also explain that total fitness effects confound selection and constraint (final paragraph of Methods, and SI section S7).

l. 706-10: rather grandiose? This is hardly a Higgs boson issue by any stretch of the imagination – not least because nothing in organismic biology allows us to make the kinds of 'strong predictions' that physics does.

I reformulated this paragraph to be clearer and less grandiose. The paragraph still contains a mention of the Higgs boson to illustrate that there are methods available that have been instrumental in other fields and that can now be used to address the problem of why human brain expansion occurred.

Robin Dunbar

Reviewer 2

Reviewer # 2:

Remarks to the Author:

I read and reviewed this paper as an evolutionary biologist interested in human brain evolution and not as a mathematician checking the equations. I can therefore contribute some general considerations that may be helpful when revising the manuscript. Overall, I find the research topic to be interesting and the findings intriguing. However, I believe that the issues mentioned below should be addressed in order to improve the clarity and accuracy of the paper.

I thank the reviewer for their helpful comments. I have sought to address them thoroughly as detailed below.

For readers with a similar background to myself, it would be helpful if the model assumptions were presented more clearly in the opening section of the paper, rather than solely in the Methods section. Additionally, it would be beneficial to specify the data input into the model more explicitly.

I have improved the introduction to better introduce the model and some key assumptions, particularly, that only females are considered and the key parameters that yield the evolution of human-scale brains. The model makes many other assumptions that are standard in evolutionary models, some of which are too technical to be described in the opening sections. For instance, the model assumes clonal reproduction, rare and weak mutation, and deterministic population dynamics. Explaining these assumptions and why they are justified is far beyond the scope of this manuscript. They have been studied extensively in many other technical papers by many authors and they are standard in particular in the field of adaptive dynamics, which is a widely used method to model phenotypic evolution.

To more explicitly specify the data input into the model, the introduction now lists the key parameter values entered into the model. It also includes references to where the empirically estimated or empirically informed parameters came from. Including this in the introduction helps highlight key parameters that should be estimated empirically to improve the model's predictions.

I have reservations about the use of point estimates for body mass and brain mass to represent entire species while ignoring within-species variance.

I agree that it would be interesting to consider the population distribution rather than only means. However, doing so would vastly complicate the model. As a first approximation and given the already high complexity introduced by considering development, the model assumes marginally small genotypic variation around the means. This is a standard assumption of adaptive dynamics, which has well known effects and has been relaxed in many evolutionary models that do not have explicit development. It will be a challenging task for the future to relax it for evolutionary models considering development.

Furthermore, it is worth noting that some data represent species averages while others represent single individuals. Even when this issue about within-group variance is disregarded, it is important to acknowledge that for most of the fossil species, estimates of body size are imprecise approximations.

Thank you for pointing this out. I included the following statement at the end of the legend of Fig. 2: "Fossil data may come from a single individual and body size estimates from fossils are subject to additional error."

I would like to stress that the model's assumption (only casually mentioned in passing) that "brain size of a newborn is fixed and cannot evolve" is unrealistic, and at odds with the well documented prenatal growth differences between humans and apes, and the (admittedly sparse) fossil evidence from hominin endocasts. I appreciate that models have to make simplifying assumptions, however, this one seems problematic.

I agree that that it is an unrealistic assumption and one that I look forward to relaxing in the near future. To facilitate finding this assumption, I now state it together with other assumptions at the start of the

Methods section. There, I also explain that it is an assumption of the evo-devo dynamics framework and a standard assumption of life history models. This assumption is made here as a first approximation and future models can build upon the work presented here to evaluate differences when the assumption is relaxed.

The term “human brain evolution” is used throughout the manuscript, but as the study includes several species that do not belong to the human genus, I would suggest using the more appropriate term “hominin brain evolution” instead. This would better reflect the scope of the study and avoid any potential confusion.

I have done as suggested and now only refer to “hominin brain expansion”, including in the title.

Regarding Box 1, I found the content to be confusing and suggest that it be removed from the manuscript entirely. While I appreciate the author’s intention to provide additional context, the information presented does not seem to add clarity or depth to the paper’s main focus.

I have done as suggested and deleted Box 1. The new Fig. 5 now presents the fitness landscape of the brain model, rather than the cartoon that was in Box 1. This new Fig. 5 more clearly shows why selection is unremarkable in the model, and why all the aspects causing human brain expansion are not captured by the fitness landscape.

In Figure 2 and the accompanying text, I recommend changing the phrase “Brain ... body size at 40 years of age” to “Adult brain and body size.” It is highly unlikely that many (if any) of the hominin individuals mentioned in this figure lived to the age of 40. Additionally, it is unclear why the authors chose to model brain size until the age of 40 specifically, so this could be explained further in the text.

I have done as suggested and now refer to “Adult brain and body size”. I also added an explanation in the second-to-last sentence of the legend of the now Fig. 3 explaining that the 40 years of age is taken as the adult age because developed traits have plateaued by that age.

In the main text you mention that “a protracted human childhood arises from the trade-off of energy allocation between brain and somatic growth, so it is a consequence of brain expansion rather than being selected for.” It would be helpful to discuss this with regards to suggestions that evidence for protracted brain growth can be observed in some Australopithecus species, which are generally considered to be rather ape-like in both endocranial volume and body mass.

I have now removed the mentioned sentence and a discussion around this point for lack of space.

was also wondering how the small endocranial volumes of *Homo naledi* and especially *Homo floresiensis* (which were excluded from the regression model, if I understood correctly) can be explained in light of the author’s model.

Our 2018 paper (González-Forero & Gardner. Inferences of ecological and social drivers of human brain-size evolution) considered *naledi* and *floresiensis*, but found that the model did not recover their brain and body sizes accurately (Extended Data Fig. 8 in that paper). This can be seen in the now Fig. 2b: the model found the same scenario as best fitting both *afarensis* and *floresiensis*, which are numbers 11 and 13 in Fig. 2b, and in such scenario the evolved brain and body sizes are given by the yellow dot approaching *P. robustus*. The scenario that generated such brain and body sizes had 90% cooperative challenges, 10% between-group competitive challenges, and strongly diminishing returns of learning (this is now illustrated in the now Fig 1, first scenario). As shown by Fig. 2b the model does not evolve exactly the brain and body sizes of *afarensis* or *floresiensis*. A likely reason may be that only a few parameters were varied (those controlling challenge proportion and the shape of EEE), suggesting that other parameter values need to be varied. The model also found a different scenario for *naledi*, with 70% cooperative challenges, 20% ecological challenges, and 10% between-group competitive challenges, but again yielding brain and body sizes that did not closely approach those of *naledi*. What this all means is that the model suggests that the brain sizes of *afarensis*, *naledi*, and *floresiensis* evolved under more different scenarios than those of the six *Homo* species studied and that to determine if the model can replicate *afarensis*,

naledi, and floresiensis it needs to explore a larger region of the parameter space.

Please explain what you are trying to accomplish by modelling “failed organisms” with random growth efforts, and which aspects of your model/assumptions you are trying to probe or test.

I have rearranged the presentation so that the paper reads more naturally. In this new arrangement, I clarified that the aim of this part of the analysis is to understand what development alone does to the developed brain and body sizes (first paragraph of “Emergence of hominin brain allometry”). This analysis considers random genotypic variation without evolution to determine which brain and body sizes develop and how they relate, finding that the brain model under the sapiens scenario has a bias toward large brains but does not produce hominin sized brains and bodies without evolution. These random genotypic traits generate “failed organisms”.

Reviewer 3

Reviewer # 3:

Remarks to the Author:

NATHUMBEHAV-23030879

Evo-devo dynamics of human brain size

This is an interesting manuscript that addresses a longstanding debate in evolutionary biology – whether social or ecological factors are primarily responsible for the evolution of the large human brain. The authors take a computational modelling approach that accounts for both evolutionary and developmental processes. I recommend major revisions to the manuscript, since clearer definitions would expand readership and the evolutionary mechanisms underlying their interpretations of the model are not clear. My comments are listed below:

I thank the reviewer for their helpful comments. I have sought to address them as much as possible.

Definitions:

What is “socio-genetic” covariation?

I included a definition in the second to last paragraph of the introduction.

The author should provide clearer definitions throughout. A figure depicting the different variables and their interactions/effects would also be helpful.

I have included definitions of additional non-standard terms, specifically, of mechanistic additive genetic co- variance in the second to last paragraph of the introduction. I have also included the new Extended Data Fig. 1 describing the causal dependencies of the brain model as suggested, which is indeed helpful.

Selection versus constraint:

This model will only be considered by evolutionary biologists if it reflects actual evolutionary mechanisms. As written, it is not clear that the scenarios presented in the text align with these mechanisms. Evolution is defined as a change in allele frequencies over generations - how exactly would the mech- anisms/relationships described enact this?

I included a sentence, the fourth in the discussion, describing the mechanism in terms of change in allele frequencies. It is “if a mutant allele coding for increased allocation to brain growth emerges, this allele can only increase in frequency in the model by being socio-genetically correlated with follicle count which is selected for, rather than due to selection for brain size.”

For instance, the author suggests that constraint can drive evolutionary change and uses Box 1 as an example. However, it appears to be that while the path represents the constraint, the volcanic eruption could represent selection.

I removed Box 1 and included a new Fig. 5 describing the actual fitness landscape of the brain model rather than the cartoon example that was present in Box 1. Additionally, hopefully the sentence just

mentioned above will clarify the evolutionary mechanism at play.

Also, the author suggests that ecology and culture drive human brain expansion in the model by affecting developmental/socio-genetic constraints rather than unconstrained selection. But how would ecological and cultural factors alter genetic covariation? Through what mechanism? There appears to be some fundamental relationship/interaction missing from this model. For instance, one major assumption upon which this model is built is that “direct selection on developmentally late reproductive tissue provides a force for reproductive tissue expansion, and socio-genetic covariation diverts this force to cause brain expansion.” How exactly would ecology and culture interact with reproductive tissue?

I included explanations for this. First, I included a new paragraph, the third in the section “Analysis of the action of selection”, which formally describes how this socio-genetic covariation arises, namely via development. Second, I included verbal descriptions of this in the abstract and discussion, stating that ecology and culture affect development and in doing so generate such covariation.

Similarly, we know that fertility is not actually proportional to the size of reproductive tissue. What could this relationship represent in true evolutionary terms?

Fertility is known to be proportional to preovulatory follicle count, which is indeed a standard clinical measure of fertility in women. I provide a citation to support this (Broekmans et al 2004). The model defines reproductive tissue as preovulatory follicles, so reproductive tissue does determine fertility in females. Perhaps confusion arose as reproductive tissue seems to include many other tissues, such as ovaries, uterus, etc. To avoid confusion, I now speak only of follicle count, measured in mass units, rather than reproductive tissue.

Minor points:

“We” versus “I” - perhaps use the latter for analyses conducted within this solo authored paper and the former for previous work by the author and their colleagues?

I now write in first person singular or passive voice. Lines

206, 214 - “consistent” instead of “consistently”

Done.

I hope the author finds my comments helpful in revising their manuscript.

Indeed, I thank the reviewer for their helpful comments.

Reviewer 4

Reviewer # 4:

Remarks to the Author:

The specific reasons behind the evolution of the human brain to its current size remain uncertain. Despite various hypotheses and comparative studies, which have provided insightful findings, they are primarily qualitative and fail to fully explain the size of the human brain. In this paper, the authors take a mathematical approach to integrate the developmental and evolutionary dynamics of human brain size. Their research reveals that the expansion of the brain is a result of the interplay between ecological factors and cultural influences, leading to covariation between brain size and the size of late-developing reproductive tissues.

This paper presents an intriguing perspective that sheds light on the understanding of the current size of the human brain. Besides, they hold great potential for elucidating the evolution of other organs, such as lungs and livers. However, I have a few suggestions to improve this manuscript.

I thank the reviewer for their helpful comments. I have sought to address them as much as possible.

1. In my view, constraints guide rather than drive evolution, with selection being the driving force. Using the example in Box 1, gravity drives the flow of mud downhill, while topographic constraints simply guide its movement.

To reduce the scope for semantic disagreement, I have removed the term “drive” everywhere. I have also removed Box 1.

2. The authors should refine the presentation of their results and consider omitting less important details. For instance, lines 126-129 mention that blue dots represent initial growth efforts, which are likely illustrated in Figure 1. Listing all 16 panels and explaining each one in the main text is unnecessary. Focusing on the main points and disregarding less relevant details or relocating them to the supplementary information (SI) section would help readers better grasp the intended message. Additionally, the use of four evolutionary time, τ , could be clarified in the caption of Figure 1 by explaining the meaning of the "horizontal axis in A."

I have removed most of the panels of the main text figures describing the evo-devo dynamics to keep only the ones with the key findings, while keeping figures with the full details in the supplementary information. This simplification of the main text figures of the evo-devo dynamics makes it easy to identify the horizontal axes and their meaning. I have also removed the excessively detailed verbal explanations of the findings.

3. Similarly, regarding the statement in lines 152-155, it is challenging to draw such a conclusion from Figure 1, which shows that “allocation to reproductive growth developmentally increases from zero after 3 years of age and slowly achieves a small maximum value at around 20 years of age.” While there may be several conclusions, the data are not particularly remarkable. The authors should avoid making such unsupported statements.

Indeed, it was not easy to see in the figure what was verbally described in those lines. I have now included insets (e.g., insets of Figs. S1e and S2e) with the image zoomed in to clearly show the described results. However, the verbal description of these results is no longer given to remove excessive detail.

4. Regarding Figure 1h, in addition to the developmental delay observed in the model compared to the actual data (red dots relative to black squares), there is also a significant difference in the developmental trend. The observed data depict rapid growth in the early stages followed by slower growth after approximately 3 years, whereas the model predicts a step-like change that lacks coherence. Similar inconsistencies can be observed in Figure 1o.

As described in the original submission and in this revision, I find that the evolved ontogenetic patterns depend on the ancestral genotypic traits. In the original submission, I used manually identified ancestral genotypic traits. In this revision, I now use the genotypic traits that evolve under the afarensis scenario (evolved from manually

identified genotypic traits) as the ancestral genotypic traits for the sapiens scenario. This removes some of the subjectivity and improves the fit of the ontogenetic patterns to observation. However, the fit is still not perfect and still differs from that found with the continuous time approach of the 2018 paper. These discrepancies are now more clearly acknowledged in the manuscript and possible reasons for the discrepancies are offered, as detailed in the response to editor and reviewers above. The final paragraph of the discussion now explains that the error in prediction relative to observation means that there is scope for improving the model or finding alternative models and that there are tools to compare such alternatives to select models best explaining the data to move toward establishing why hominin brain expansion happened using this approach.

7th December 2023

Dear Dr González-Forero,

Thank you once again for your manuscript, entitled "Evo-devo dynamics of hominin brain size," and for your patience during the peer review process.

Your revised manuscript has now been evaluated by Reviewers 3 and 4 from the previous round, whose comments are included at the end of this letter (and in the code review document, attached). Although the reviewers find your work to have improved in revision, they also raise some important outstanding concerns. We are very interested in the possibility of publishing your study in *Nature Human Behaviour*, but would like to consider your response to these concerns in the form of a revised manuscript before we make a decision on publication.

In particular, we ask that you

1) Edit your code to ensure that it accurately reproduces the figures that appear in your manuscript, is free from errors and mentions all packages that are used. We strongly encourage you to follow Reviewer 4's suggestion to create a dedicated document for each figure and ensure file output names clearly identify the figure they correspond to.

2) Edit the framing of your findings (in both your Abstract and main text) to address the concerns raised by Reviewer 3, to ensure that the contribution of the work is clearly and accurately stated.

In sum, we invite you to revise your manuscript taking into account all reviewer and editor comments. We are committed to providing a fair and constructive peer-review process. Do not hesitate to contact us if there are specific requests from the reviewers that you believe are technically impossible or unlikely to yield a meaningful outcome.

We hope to receive your revised manuscript within two months. I would be grateful if you could contact us as soon as possible if you foresee difficulties with meeting this target resubmission date.

- Include a "Response to the editors and reviewers" document detailing, point-by-point, how you addressed each editor and referee comment. If no action was taken to address a point, you must provide a compelling argument. When formatting this document, please respond to each reviewer comment individually, including the full text of the reviewer comment verbatim followed by your response to the individual point. This response will be used by the editors to evaluate your revision and sent back to the reviewers along with the revised manuscript.
- Highlight all changes made to your manuscript or provide us with a version that tracks changes.

[REDACTED]

Note: This URL links to your confidential home page and associated information about manuscripts you may have submitted, or that you are reviewing for us. If you wish to forward this email to co-authors,

please delete the link to your homepage.

We look forward to seeing the revised manuscript and thank you for the opportunity to review your work. Please do not hesitate to contact me if you have any questions or would like to discuss these revisions further.

Sincerely,

[REDACTED]

REVIEWER COMMENTS:

Reviewer #3:

Remarks to the Author:

NATHUMBEHAV-23030879

Evo-devo dynamics of hominin brain size

I very much appreciate the author's efforts to address Reviewer concerns.

However, I still have an issue with how the results are presented. Namely, in the abstract (and throughout the manuscript) it is stated that "in this model the brain expands because it is 'socio-genetically' correlated with developmentally late preovulatory ovarian follicles, not because brain size is directly selected for" – this statement is likely to be misinterpreted as a finding gleaned from some modelling executed in the manuscript (for example, from model likelihood comparisons) rather than a necessary result of the model structure and assumptions. In particular, follicle count is the only factor related to fitness (Extended Data Fig 1), so this necessarily means that selection can't directly occur on any other phenotypes, including brain size. This needs to be reframed so that readers don't incorrectly interpret what this modelling framework is adding to the literature on brain size evolution.

Reviewer #4:

Remarks to the Author:

The author has addressed all of my previous concerns. There was clearly a lot of effort put into this revision, and I think it has considerably improved the paper. In principle, I can recommend it for acceptance.

[please see attachment for Reviewer 4's code review]

Comments about the codes

1. All codes were executed successfully, yielding numerous plots. However, Fig. 2b is the only one consistent with the manuscript; Fig. 1 lacks representation of model results, as the codes generate datasets for seven hominin species but the plot only displays empirical data. A thorough review of these codes is recommended to address this discrepancy.

2. To enhance reproducibility, consider creating a dedicated document for each figure. This approach simplifies result replication by enabling users to obtain each figure through the execution of a single file. The current process of substituting different parameter sets is intricate and could benefit from a more straightforward approach.
3. File output names should be straightforward and easily identifiable, such as Fig. 2a, Fig. 2b, SFig. 3a, etc. Incorporating parameter sets in the file names complicates locating the corresponding figures in both the manuscript and supplementary information.
4. It is advisable to conduct a thorough code cleanup, removing unnecessary lines like the specific file path `"cd(); cd("Documents/Academia/Postdoc/2StAndrews/Papers/8DiscreteBrain/Manuscript5Brain/Runs")"`, which have been identified as causing errors.
5. In addition to specifying the installation requirements for Julia and Visual Studio Code, it is crucial to mention the installation of essential packages such as CSV and DataFrames to ensure a seamless setup for users attempting to replicate the results.
6. Provide information on the time required to obtain results. For instance, it took approximately 20 minutes to generate results for each hominin species in Fig. 1. This

inclusion aids users in managing expectations regarding the computational resources and time investment necessary for result reproduction.

Author Rebuttal, first revision:

Reviewer 3

Reviewer # 3: Remarks to the Author: NATHUMBEHAV-23030879 Evo-devo dynamics of hominin brain size

I very much appreciate the author's efforts to address Reviewer concerns.

However, I still have an issue with how the results are presented. Namely, in the abstract (and through- out the manuscript) it is stated that "in this model the brain expands because it is 'socio-genetically' correlated with developmentally late preovulatory ovarian follicles, not because brain size is directly selected for" – this statement is likely to be misinterpreted as a finding gleaned from some modelling executed in the manuscript (for example, from model likelihood comparisons) rather than a necessary result of the model structure and assumptions. In particular, follicle count is the only factor related to fitness (Extended Data Fig 1), so this necessarily means that selection can't directly occur on any other phenotypes, including brain size. This needs to be reframed so that readers don't incorrectly interpret what this modelling framework is adding to the literature on brain size evolution.

I have made some slight but important changes to clarify this point. The fact that, in the model, selection cannot directly act on any phenotype other than follicle count is an insight gained from the deeper analysis enabled by the evo-devo dynamics framework used here, rather than an insight that was already clear from previous work. This point was not clear as suggested by the reviewer's comment because I referred to the Extended Data Fig. 1 very early in the introduction when describing the model (line 79 in the previous submission), which suggested that such insight was already available from the model construction. To clarify that this insight was not available, I moved my mention of the Extended Data Fig. 1 to later in the introduction when describing the insights gained from this paper (line 155 in the current submission). To further clarify this, I included statements that the insight that there is no direct selection for any phenotype other than follicle count is drawn from the present analysis (lines 703-705 and first two lines of the legend of Extended Data Fig. 1).

Regarding the suggested misunderstanding that conclusions are obtained from likelihood comparisons, I have carefully referred to "this model" throughout, including in the abstract and in the quoted sentence, and devote the final paragraph of the discussion to explain that no other model was considered so likelihood comparisons between models is not in the scope of this paper. That same paragraph explains that the computational speed achieved now with this submission makes it feasible to carry out such model comparisons in the future.

Reviewer 4

Reviewer # 4: Remarks to the Author: The author has addressed all of my previous concerns. There was clearly a lot of effort put into this revision, and I think it has considerably improved the paper. In principle, I can recommend it for acceptance.

[please see attachment for Reviewer 4's code review]

I thank the reviewer for evaluating the manuscript again and for their careful review of the code. The reviewer's comments allowed me to greatly improve the code and its reproducibility.

1. All codes were executed successfully, yielding numerous plots. However, Fig. 2b is the only one consistent with the manuscript; Fig. 1 lacks representation of model results, as the codes generate datasets for seven hominin species but the plot only displays empirical data. A thorough review of these codes is recommended to address this discrepancy.

I thoroughly reviewed the code as suggested. Among other things, the reviewed code solves the discrepancy found by the reviewer and generates each figure by simply clicking a button.

2. To enhance reproducibility, consider creating a dedicated document for each figure. This approach simplifies result replication by enabling users to obtain each figure through the execution of a single file. The current process of substituting different parameter sets is intricate and could benefit from a more straightforward approach.

Thank you. I have done as requested, so there is now a dedicated file for each figure. This makes it straightforward to run the model and reproduce its results.

3. File output names should be straightforward and easily identifiable, such as Fig. 2a, Fig. 2b, SFig. 3a, etc. Incorporating parameter sets in the file names complicates locating the corresponding figures in both the manuscript and supplementary information.

Done.

4. It is advisable to conduct a thorough code cleanup, removing unnecessary lines like the specific file path `"cd(); cd("Documents/Academia/Postdoc/2StAndrews/Papers/8DiscreteBrain/Manuscript5Brain/Runs")"`, which have been identified as causing errors.

Done. I thoroughly cleaned the code and among other things replaced the code line quoted by a generic line that yields no error in other computers.

5. In addition to specifying the installation requirements for Julia and Visual Studio Code, it is crucial to mention the installation of essential packages such as CSV and

DataFrames to ensure a seamless setup for users attempting to replicate the results.

Done.

6. Provide information on the time required to obtain results. For instance, it took approximately 20 minutes to generate results for each hominin species in Fig. 1. This inclusion aids users in managing expectations regarding the computational resources and time investment necessary for result reproduction.

I have now included detailed information on the time required to obtain results. This information is included in the readme file as well as appearing in the terminal when running the code.

Regarding running time, I find that it takes about 4 min to generate results for each hominin species, rather than the 20 min found by the reviewer. This extended time found by the reviewer may have been due to the previous intricate instructions to manually enter parameter values, which may have led the reviewer to enter incorrect values that made the model ill-specified and so substantially slower yielding the discrepant results found by the reviewer. I have now fixed this in the revised code, as the results are recovered by simply pressing a button for the dedicated files for each figure without manually entering parameter values.

Decision Letter, second revision:

11th March 2024

Dear Dr. González-Forero,

Thank you for your patience as we've prepared the guidelines for final submission of your Nature Human Behaviour manuscript, "Evo-devo dynamics of hominin brain size" (NATHUMBEHAV-23030879B). Please carefully follow the step-by-step instructions provided in the attached file, and add a response in each row of the table to indicate the changes that you have made. Please also check and comment on any additional marked-up edits we have proposed within the text. Ensuring that each point is addressed will help to ensure that your revised manuscript can be swiftly handed over to our production team.

We would hope to receive your revised paper, with all of the requested files and forms within two-three weeks. Please get in contact with us if you anticipate delays.

Nature Human Behaviour offers a Transparent Peer Review option for new original research manuscripts submitted after December 1st, 2019. As part of this initiative, we encourage our authors to support increased transparency into the peer review process by agreeing to have the reviewer comments, author rebuttal letters, and editorial decision letters published as a Supplementary item. When you submit your final files please clearly state in your cover letter whether or not you would like to participate in this initiative. Please note that failure to state your preference will result in delays in accepting your manuscript for publication.

In recognition of the time and expertise our reviewers provide to Nature Human Behaviour's editorial process, we would like to formally acknowledge their contribution to the external peer review of your manuscript entitled "Evo-devo dynamics of hominin brain size". For those reviewers who give their assent, we will be publishing their names alongside the published article.

Cover suggestions

We welcome submissions of artwork for consideration for our cover. For more information, please see our guide for cover artwork.

ORCID

Non-corresponding authors do not have to link their ORCIDs but are encouraged to do so. Please note that it will not be possible to add/modify ORCIDs at proof. Thus, please let your co-authors know that if they wish to have their ORCID added to the paper they must follow the procedure described in the following link prior to acceptance:

Nature Human Behaviour has now transitioned to a unified Rights Collection system which will allow our Author Services team to quickly and easily collect the rights and permissions required to publish your work. Approximately 10 days after your paper is formally accepted, you will receive an email in providing you with a link to complete the grant of rights. If your paper is eligible for Open Access, our Author Services team will also be in touch regarding any additional information that may be required to arrange payment for your article.

Please note that *Nature Human Behaviour* is a Transformative Journal (TJ). Authors may publish their research with us through the traditional subscription access route or make their paper immediately open access through payment of an article-processing charge (APC). Authors will not be required to make a final decision about access to their article until it has been accepted. Find out more about Transformative Journals

[REDACTED]

Best regards,
[REDACTED]

On behalf of

[REDACTED]

Reviewer #4:

Remarks to the Author:

The author has addressed all of my concerns regarding the codes. I can confirm that all figures presented in the paper can be replicated by executing the provided codes. Based on this thorough evaluation, I recommend the acceptance of this paper.

Final Decision Letter:

Dear Dr González-Forero,

We are pleased to inform you that your Article "Evo-devo dynamics of hominin brain size", has now been accepted for publication in *Nature Human Behaviour*.

Please note that *Nature Human Behaviour* is a Transformative Journal (TJ). Authors may publish their research with us through the traditional subscription access route or make their paper immediately open access through payment of an article-processing charge (APC). Authors will not be required to make a final decision about access to their article until it has been accepted. Find out more about Transformative Journals

With best regards,
[REDACTED]